



# Methodological and physical biases in global to sub-continental borehole temperature reconstructions: an assessment from a pseudo-proxy perspective

Camilo Melo-Aguilar [1,2], J. Fidel González-Rouco [1,2], Elena García-Bustamante [3], Norman Steinert [1,2], Johann H. Jungclaus [4], Jorge Navarro [3], and Pedro J. Roldán-Gómez [1]

[1]Universidad Complutense de Madrid, 28040 Madrid, Spain
[2]Instituto de Geociencias, Consejo Superior de Investigaciones Cientificas-Universidad Complutense de Madrid, 28040 Madrid, Spain
[3]Centro de Investigaciones Energéticas, Medioambientales y Tecnológicas (CIEMAT), 28040 Madrid, Spain
[4]Max Planck Institut für Meteorologie, Hamburg, Germany
**Correspondence:** Camilo Melo-Aguilar (camelo@ucm.es)

**Abstract.** Borehole-based reconstruction is a well-established technique to recover information of the past climate variability based on two main hypothesis: first, that past ground surface temperature (GST) histories can be recovered from borehole temperature profiles (BTPs); and second, that the past GST evolution is coupled to surface air temperature (SAT) changes and thus, past SAT changes can be recovered from BTPs. Compared to some of the last millennium (LM) proxy-based reconstruc-

tions, previous studies based on the borehole technique indicate a larger temperature increase during the last centuries. The nature of these differences has fostered the assessments of this reconstruction approach searching for potential causes of bias. Here, we expand previous works to explore potential methodological and physical bias using pseudo-proxy experiments with the Community Earth System Model-Last Millennium Ensemble (CESM-LME). A heat-conduction forward model driven by simulated surface temperature is used to generate synthetic BTPs that are then inverted using singular value decomposition.

This procedure is applied to the set of simulations that incorporate all the LM external forcing factors as well as those that consider the concentration of the green house gases (GHG) and the land use land cover (LULC) changes forcings separately. The results indicate that methodological issues may impact the representation of the simulated GST at different spatial scales, with the temporal logging of the BTPs as the main sampling issue that may lead to an underestimation of the simulated GST 20th century trends. Our analysis also shows that in the surrogate reality of the CESM-LME the GST does not fully capture

the SAT warming during the industrial period and thus, there may be a further underestimation of the past SAT changes due to physical processes. Globally, this effect is mainly influenced by the GHG forcing whereas regionally, LULC changes and other forcings factors also contribute. These findings suggest that despite the larger temperature increase suggested by the borehole estimations during the last centuries of the LM relative to some other proxy reconstructions, both the methodological and physical biases would result in a underestimation of the 20th century warming.



## 1  Introduction

Over pre-instrumental times, the climate evolution is estimated from a variety of indirect sources used as proxy indicators of climate variations (Houghton et al., 2001). During the last decades, there have been advances in expanding and improving proxy-temperature reconstructions targeting the common era (CE; Masson-Delmotte et al., 2013). This information offers a frame for addressing the 20th century warming in a broader temporal context, contributing to a more suitable analysis of the forced climate system response with respect to the natural background. Overall, different hemispheric to global scale proxy-reconstructions depict similar climate variability during the last 2000 years. However, the amplitude and timing of past climate states are still not fully constrained. For instance, the magnitude of the temperature variation from the Little Ice Age (LIA; ~1450-1750 CE) to present day displays a range of uncertainty in the estimations of different proxy-sources. While some of them estimate a temperature increase of ~0.6°C (e.g. Ammann and Wahl, 2007), other sources suggest a larger warming of ~1 to ~1.4°C (e.g. Pollack and Smerdon, 2004; Frank et al., 2007). In general, there are uncertainties regarding the amplitude of low-frequency changes within the Last Millennium (LM) as in the transition from the Medieval Climate Anomaly (MCA; ~0950-1250 CE). The magnitude of such changes depends on the climate sensitivity of the system and it is important that reconstructions and climate models are consistent on its estimation (Fernández-Donado et al., 2013). Therefore, assessing the origin and levels of these uncertainties in the amplitude and time of the temperature changes is pivotal in the context of LM climate variations.

One of the proxy reconstructions yielding a larger warming from the LIA to present is that arising from the borehole technique. This method is based on the assumption that the soil captures the surface climatic signal due to the conductive heat transport of the ground surface temperature (GST) variations into the subsurface. Therefore, the inversion of borehole temperature profiles (BTPs) provides an estimation of the past GST variations (Pollack and Huang, 2000). In addition, it is also assumed that the surface air temperature (SAT) is strongly coupled to GST (Smerdon et al., 2004), and thus, borehole reconstructions stand as a good proxy of the past SAT variation. Due to the conductive propagation of the temperature signal with depth, only the low-frequency variations are recorded in the subsurface (Pollack and Huang, 2000) and consequently, the inversion of BTPs can offer an estimation of centennial long term trends.

The borehole reconstructed temperatures at hemispheric scales have been subject to assessment and debate during the last decades due to their estimated multi-centennial trend of ca. 1°C over the 1500-2000 CE period, a relatively large value in comparison with other proxy sources (Jones et al., 2009; Fernández-Donado et al., 2013; Masson-Delmotte et al., 2013). It has been discussed whether some methodological limitations and environmental or physical factors may somewhat hinder achieving robust borehole based estimations of past temperature trends (e.g. Rutherford and Mann, 2004; Pollack and Smerdon, 2004; González-Rouco et al., 2009). Methodological aspects refer to a variety of issues that encompass (Pollack and Huang, 2000; Bodri and Cermak, 2007): site specific processes contributing to noise in BTPs, like interactions with orography and hydrology leading to horizontal (vertical) advection (convection) that render the conductive regime assumption invalid; and sampling irregularities such as low density and/or irregular spatial distribution of boreholes at regional and larger spatial scales, regional differences and variability in logging dates and depth of profiles, vertical resolution, errors in measurements,



etc. All these methodological issues may have an impact on the recovery of past GSTs from BTPs. In turn, changes in surface

environmental or boundary conditions may have an impact on the recovery of past SAT variations from reconstructed GSTs, i.e. on SAT-GST coupling. Long term variability in surface climate parameters like changes in snow cover, freezing, land use and land cover (LULC) or evaporation changes may therefore play a role in the surface energy balance and thus in SAT-GST interactions (Mann and Schmidt, 2003; Smerdon et al., 2004; González-Rouco et al., 2003, 2009).

    The impact of methodological issues on hemispherical scale reconstructions, like the uneven geographical distribution of the

borehole sites and the spatial gridding of the data where discussed by Rutherford and Mann (2004) and Pollack and Smerdon (2004) using the observational borehole dataset (Huang and Pollack, 1998). They showed that the spatial aggregation and weighting scheme, used in Huang et al. (2000) and Harris and Chapman (2001) has minor implications for the warming trends indicated by these reconstructions. They also argued that the predominantly mid-latitude distribution of boreholes should be able to capture the northern hemisphere (NH) temperature evolution. Likewise, Beltrami and Bourlon (2004) addressed the

spatial aggregation of the borehole sites to obtain NH averages, using gridding instead of geographic aggregation and various grid cell sizes. They reported a 1°C NH GST increase for the 1500-1980 CE period consistent with the Huang et al. (2000) and Harris and Chapman (2001) estimations. Additional sampling issues like differences in timing and depth of borehole logs have received less attention. Most of the logs were done before 1980 CE. Thus, the aggregation of this information should be done with caution as the post 1980 years, when global warming has been larger, are underrepresented. Their influence is thus

difficult to assess in studies with real BTPs. Huang et al. (2000) used fixed century-long ramp trends for each BTP to estimate hemispheric long-term temperature variations from 1500-2000 CE. The estimated trend does not fully capture the warming within the last decades of the 20th century, as the bulk of the borehole sites were logged before 1990 CE. Harris and Chapman (2001) reconstructed the 1500-2000 CE NH mid-latitude temperatures from a set of BTPs that were taken to a common time frame. They forward propagated each reduced temperature profile by considering a constant temperature evolution from the

logging date up to 1995 CE. Both of these analyses represent a partially muted estimation of the industrial warming since only an small portion of BTPs data include information of the last decades of the 20th century (Pollack and Smerdon, 2004; Jones et al., 2009). Both works yield a similar NH GST increase, and thus an equivalent SAT increase considering SAT-GST coupling, of about ~1°C for the 1500-2000 CE interval.

    The sensitivity of the SAT-GST coupling to some physical processes at the surface has also been addressed. The discussion

has been focused on whether variations in the long-term behavior of surface properties may result in a biased representation of the SAT histories by the reconstructed GST (Jansen et al., 2007). This thread of analysis has benefited considerably from introducing contributions from modeling. For instance, the influence of snow cover was analyzed by various studies under the hypothesis that changes in snow cover may influence the SAT-GST offset and produce long term drifts in air-ground coupling. Bartlett et al. (2005) studied this issue in multidecadal observational records and several studies (Mann and Schmidt, 2003;

Chapman et al., 2004; Schmidt and Mann, 2004) used 50-yr long simulations with the GISS ModelE general circulation model (GCM). Their results suggest that hemispheric scale reconstructions are unlikely to suffer from snow cover biases. Nevertheless, long term changes in snow cover may alter SAT-GST tracking by introducing transient and persistent long term signatures in the coupling, for example by virtue of changes in external forcings. González-Rouco et al. (2003, 2009)




evaluated this at multi-centennial timescale by considering millennial-long simulations of the coupled atmosphere-ocean GCM

ECHO-G (Legutke and Voss, 1999) driven by both natural (solar and volcanic) and anthropogenic (greenhouse gases; GHG) forcing factors. They found that in spite of existing decadal variability SAT-GST coupling was reasonably stable at global and hemispheric scales, thus supporting the use of BTPs at those scales. At regional scales, snow cover trends may develop that compromise BTP based reconstructions.

Other issues that may play a role in disrupting SAT-GST coupling through changes in the energy balance and that have

received some attention are soil moisture variations (González-Rouco et al., 2009; Cermak and Bodri, 2018) and LULC changes (Cermak et al., 2017; MacDougall and Beltrami, 2017). Melo-Aguilar et al. (2018) addressed these issues by considering for the first time SAT-GST coupling in a large ensemble of simulations of the LM (CESM-LME; Otto-Bliesner et al., 2016) that include a sub-ensemble of experiments involving a realistic set up of all-forcing natural (solar, volcanic and orbital) and anthropogenic (GHG, anthropogenic aerosols and LULC changes) as well as setups of specific single-forcing sub-ensembles.

This allowed for separating the effect of each forcing and understanding the integral effects when all forcings were used from the single forcing contributions and feedbacks. At global and hemispheric scales SAT-GST coupling still holds in the all-forcings simulations as a result of compensating effects of various forcings contributing to energy balance. At regional scales SAT-GST coupling is compromised by the influence of specific forcings, and often amplified by changes in snow cover. This is relevant for global and continental BTP based reconstructions because spatial sampling of BTPs should be representative of

the target signal and avoid locations where SAT-GST become progressively decoupled.

An evaluation of the impacts of methodological and physical issues on borehole based reconstructions can be done by considering model simulations as a surrogate reality for the actual climate evolution (Smerdon, 2012). The simulations are considered as physically plausible climate realizations, compatible with the external forcings imposed and complex enough to allow for a credible implementation of the reconstruction method. The use of millennium-length GCM simulations has provided

a long-term framework to analyze the physical background of the SAT-GST relationship and has allowed for developing such pseudo-proxy experiments (PPEs). For instance, González-Rouco et al. (2006) used LM simulations (1000-1990 CE) from the ECHO-G model in a PPE. They used simulated GSTs and a forward model to simulate BTPs of 600 m depth. These were subsequently inverted following standard procedures in borehole reconstruction strategies and obtained a low pass filtered version of SAT long term trends that could be afterwards compared to the simulated SATs for verification. Thus, they found

that the method could appropriately retrieve past SAT long term trends. Additionally to this idealized case in which BTPs would be available for every model gridpoint, they also implemented the method in a more realistic case by limiting the sampling of BTPs to replicate their actual distribution in reality. This exercise rendered similar robust results. González-Rouco et al. (2009) extended this analysis in order to include the effects of the variability in the timing of BTP logging dates and their depths for a specific example over North America. Their results suggest that such variability does not prevent the borehole technique from

retrieving the North American 20th century warming. García-García et al. (2016) used the LM all-forcing simulations from state-of-the-art Earth System Models (ESMs) within the frame of the CMIP5/PMIP3 (Coupled Model Intercomparison Project phase 5 / Paleoclimate Modeling Intercomparison Project phase 3; Taylor et al., 2012). They followed the idealized approach



in González-Rouco et al. (2006) sampling the full model grid over land. This allowed for demonstrating the performance of the borehole method over the current generation of ESMs, involving different land models and surface parameterizations.

The previous studies indicate that, globally, GST should be a good proxy of the past long-term SAT variations. Additionally, they support the overall performance of the borehole method at hemispheric and global scales under realistic scenarios involving a full setup of PMIP3 natural and anthropogenic forcings (Schmidt et al., 2011, 2012). In this work, we elaborate from previous analyses and present a set of PPEs in which we implement a borehole reconstruction strategy using LM simulations of a ESM and assess reconstructions performance for a range of spatial scales. We update analyses of methodological and physical

influences on GST reconstructions and SAT-GST coupling, respectively, that had not been systematically addressed before. We build on the analysis of Melo-Aguilar et al. (2018) by using the same NCAR coupled ESM LM ensemble (CESM-LME hereafter) and design PPEs to evaluate the influence of sampling methodological issues at global and regional scales by considering the actual distribution of BTPs as well as their depths and logging times. Additionally, we use a sub-ensemble of simulations incorporating either full setup of natural and anthropogenic forcings, the so-called all-forcing experiments (ALL-*F* hereafter),

and sub-ensembles of experiments incorporating specific individual forcings, the so-called single-forcing experiments (single-*F* hereafter). This allow for gaining understanding of how forcing factors and their influence on SAT-GST coupling can impact the results on different regions. Particularly, we focus on the GHG and LULC only (GHG-only and LULC-only hereafter) ensembles due their highest potential for impacting the SAT-GST relationship (Melo-Aguilar et al., 2018).

    In the first part of the manuscript (Sect. 2), the general characteristics of the model and the simulations employed in this

study are presented. Subsequently, Sect. 3 describes the pseudo-proxy configuration considered herein. Results are described in Sect. 4, including the specific effects due to the methodological (Sect. 4.1) and the physical issues (Sect. 4.2). The latter considers also independently the contribution of the LULC and GHG external forcings. Finally, Sect. 5 wraps up conclusions and discussion of the main results.

## 2   ESM simulations

CESM-LME simulations covering the period 850-2005 CE produced with the Community Earth System Model version 1.1 (CESM1; Hurrell et al., 2013), are used. The CESM1 includes the Atmosphere Model version 5 (Neale et al., 2012) and the Parallel Ocean Program version 2 (Smith et al., 2010) as well as the Los Alamos sea ice model (Hunke et al., 2015). The CESM-LME has a horizontal resolution of ~2° over the atmosphere and land, and ~1° in the ocean and sea ice areas.

    The Community Land Model version 4 (CLM4; Lawrence et al., 2011) represents the land surface component in the CESM1.

One of the main characteristics of the CLM4 is that the bottom boundary condition placement (BBCP), at 42.1 m depth, is the deepest among the current generation of land surface models with a soil column discretized into 15 layers (Table 1) including up to 5 additional layers in the overlying snowpack. In this respect, the CLM4 includes some improvements in the representation of some surface processes, relative to previous versions and arguably to other models (García-García et al., 2019). These include a better description of ground evaporation, thermal and hydrology properties of organic soils, snow albedo, snow cover

fraction and burial fraction of vegetation by snow (Oleson et al., 2010). Such features make the CLM4 specifically suited for



the purpose of this work as they allow for a state-of.the-art representation of the energy transfer between the atmosphere and the soil, a key aspect in the SAT-GST relationship. Further, the relatively deep BBCP allows for a better representation of the downward propagation of the temperature signal at longer (e.g. decadal and centennial) timescales (Alexeev et al., 2007; Stevens et al., 2007; MacDougall et al., 2008).

**Table 1.** Soil layers and node depths in CLM4. The node depth, which indicates the depth where the thermal properties are defined for soil layers (Oleson et al., 2010), does not necessarily coincide with the center of the layer depth.

| Layer | Layer depth (m) | Node depth (m) |
|-------|-----------------|----------------|
| L1 | 0.017 | 0.007 |
| L2 | 0.045 | 0.027 |
| L3 | 0.090 | 0.062 |
| L4 | 0.165 | 0.118 |
| L5 | 0.289 | 0.212 |
| L6 | 0.492 | 0.366 |
| L7 | 0.828 | 0.619 |
| L8 | 1.382 | 1.038 |
| L9 | 2.296 | 1.727 |
| L10 | 3.801 | 2.864 |
| L11 | 6.284 | 4.739 |
| L12 | 10.377 | 7.829 |
| L13 | 17.125 | 12.925 |
| L14 | 28.252 | 21.326 |
| L15 | 42.103 | 35.177 |

The CESM-LME includes an ensemble of ALL-*F* simulations that incorporates the complete set of agreed CMIP5/PMIP3 LM external forcings (Schmidt et al., 2011, 2012), including the natural and anthropogenic components. Up to date, a total of 13 ALL-*F* simulations are available. Additionally, smaller ensembles of single-forcing simulations, that consider each of the LM external forcings separately, are included (see Table 2 for details and references therein). This work makes use of the ALL-*F* ensemble as well as the GHG- and the LULC-only ensembles. Table 3 presents a detailed description of the original
model output as it is described in Otto-Bliesner et al. (2016).





**Table 2.** External forcing reconstructions used in the CESM-LME, both in the ALL-*F* and the single-*F* sub-ensembles. Legend for external forcing: SOL, changes in total solar irradiance; VOLC, volcanic activity; GHG, concentrations of the well-mixed greenhouse gases $CO_2$, $CH_2$, and $N_2O$; LULC, land use land cover changes; ORB, orbital variations; and OZ/AER, anthropogenic ozone and aerosols.

| Forcing | Reference |
|---|---|
| ALL-*F* | - |
| SOL | Vieira et al. (2011) |
| VOLC | Gao et al. (2008) |
| GHG | MacFarling Meure et al. (2006) |
| LULC | Pongratz et al. (2008) dataset, spliced to Hurtt et al. (2009) at 1500 CE. The only plant functional types (PFTs) that are changed are those for crops and pasture; all other PFTs remain at their 1850 control prescriptions |
| ORB | The CESM model adjusts yearly orbital position (eccentricity, obliquity and precession) following Berger et al. (1993) |
| OZ/AER | Fixed at the 1850 control simulation values until 1850 CE when the evolving anthropogenic changes up to 2005 CE. Stratospheric aerosols are prescribed in the model as a fixed single-size distribution in the three layers in the lower stratosphere above the tropopause. The ozone forcing is from the Whole Atmosphere Community Climate Model (WACCM) |

**Table 3.** CESM-LME simulations used in this study. The first and second columns present the acronyms used in this manuscript for the ensembles and the ensemble members, respectively. The ID of the original experiment files is provided in column 3.

| Ensemble acronym | Ensemble member | Simulation id |
|---|---|---|
| ALL-*F* | ALL-F$_i$ | b.e11.BLMTRC5CN.f19_g16.0i |
| | i=1,...,13 | i=01,...,13 |
| GHG-only | GHG$_i$ | b.e11.BLMTRC5CN.f19_g16.GHG.00i |
| | i=1,2,3 | i=1,2,3 |
| LULC-only | LULC$_i$ | b.e11.BLMTRC5CN.f19_g16.LULC_HurttPogratz.00i |
| | i=1,2,3 | i=1,2,3 |





## 3 Experimental design

The theoretical basis for the borehole temperature reconstruction states that the subsurface contains a thermal signature of the past surface temperature variations due to the superposition of the downward temperature signal propagating onto the background geothermal gradient. Therefore, the inversion of BTPs can yield information of the past surface temperature changes.
The information of the last 500 to 1000 years is retained within the upper few hundred meters of the subsurface (Beltrami and Bourlon, 2004). Thus, BTPs of at least 300 m depth are required to retrieve the past 500 years (Jaume-Santero et al., 2016) whereas deeper profiles of at least 500 m are necessary to retrieve information of the complete LM (Pollack and Huang, 2000).

Within the model world, the depth of BTPs is limited to the land model depth. Although the CLM4 has the deepest BBCP among the current land surface models, it is still too shallow to directly provide such deep BTPs to account for LM temperature
variations (see Table 1) . Consequently, these profiles must be synthetically generated from the surrogate reality of the simulated world as in González-Rouco et al. (2006). The following section provides an overall description of the models employed in this study to simulate the synthetic BTPs as well as for the inversion of the resulting profiles. Further details can be found in Mareschal and Beltrami (1992).

### 3.1 Forward and inverse models

The temperature at any depth $z$ is determined by the combination of the geothermal heat flux and the temperature perturbation $T_t(z)$ induced by the surface temperature variations:

$$T(z) = T_0 + q_0 R(z) + T_t(z) \tag{1}$$

where $q_0$ represents the surface heat flow density, $R(z)$ is the thermal depth and $T_0$ is a reference ground temperature. In the forward model, $T_0 + q_0 R(z)$ can be set equal to 0 because the aim is to derive $T(z)$ only as a function of the past surface
temperature variations. The forward model, thus, determines the transient perturbation component $T_t(z)$.

The propagation of the surface temperature signal within the subsurface is controlled by the one-dimensional time-dependent heat conduction equation (Carslaw and Jaeger, 1959):

$$\frac{\delta T}{\delta t} = \kappa \frac{\delta T}{\delta z} \tag{2}$$

with $\kappa$ as the thermal diffusivity and $z$ and $T$ as depth and temperature, respectively. Equation (2) is solved for an instantaneous
temperature change at time $t$ before present as:

$$T_t(z) = \Delta T \, erfc\left(\frac{z}{2\sqrt{\kappa t}}\right) \tag{3}$$

where $erfc$ is the complementary error function. $T_t(z)$ can be modeled by considering the temperature variations at the surface as a series of $K$ time step temperature changes. In this way, each step imprints a thermal signature in the subsurface that is merged to the signature of the previous step. Thus, $T_t(z)$ is given by (Mareschal and Beltrami, 1992):

$$T_t(z) = \sum_{k=1}^{K} \Delta T_k \left[ erfc\left(\frac{z}{2\sqrt{\kappa t_k}}\right) - erfc\left(\frac{z}{2\sqrt{\kappa t_{k-1}}}\right) \right] \tag{4}$$





where $\Delta T_k$ are the surface temperature changes for $K$ time intervals, each value representing an average over time $(t_k - t_{k-1})$.

Equation (4) is used to create synthetic BTPs, using LM surface temperature annual anomalies from the CESM-LME as the upper boundary condition. $T_t(z)$ is evaluated at every 1 m depth interval up to a depth of 600 m in order to accommodate for the propagation of the LM surface temperature variations. The thermal diffusivity used in the geothermal models is $1.5 \times 10^{-6}$ m$^2$ s$^{-1}$, obtained from the values of the bedrock thermal conductivity ($3.0$ W m$^{-1}$ K$^{-1}$) and volumetric heat capacity ($2 \times 10^6$ J m$^{-3}$ K$^{-1}$) of the CLM4 (Lawrence et al., 2011).

For the inversion of the synthetic BTPs, singular value decomposition (SVD; Mareschal and Beltrami, 1992) is applied to retrieve the past long-term surface temperature variations. For the present work, the inverse model consists of a series of 15-yr step changes in surface temperature histories following García-García et al. (2016). Different parameterizations were also tested (e.g. 20-yr or 25-yr step changes; not shown) and showed consistent results with the 15-yr discretization. The latter was finally selected because it yielded a convinient representation of the GTS histories. Likewise, the number of retained singular values is also important. The presence of small singular values leads to an unstable solution dominated by noise while the use of only a few principal components (PCs) result in a smoothed low resolution solution (Beltrami and Bourlon, 2004). The selection of the number of singular values is done by setting an eigenvalue cutoff, from which smaller values are eliminated. In this study we have used a cutoff value of $1.5 \times 10^{-1}$ from which the solution is the linear combination of the three leading modes. We have additionally explored the effect on the solution of retaining four and five PCs.

## 3.2 Pseudo-proxy set up

The assessment of the impacts of the methodological and physical issues described above on borehole temperature reconstructions, is done by designing two pseudo-proxy configurations. First, a so-called ideal borehole scenario (IBS) is considered, in which a BTP is simulated at every land model grid point up to a depth of 600 m. This scenario is also characterized by a homogeneous logging date at the end of the CESM-LME simulation period (2005 CE) at every grid point. The latter is achieved by driving the forward model with the annual temperature anomalies calculated with respect to the 850-2005 CE mean. Subsequently, each of these BTPs is inverted and a latitude weighted average is used to obtain the global mean. A similar approach has been used in previous works showing that under such idealized configuration, the borehole technique is able to retrieve the simulated surface temperature variations at global scale (González-Rouco et al., 2006; García-García et al., 2016). Hence, this scenario provides a benchmark experiment that will allow for evaluating the impacts of having some methodological constraints that mimic those in reality. The IBS scenario embraces two cases. On the one hand, the model soil temperature (ST) at layer 12 (ST$_{L12}$ ~7.8 m depth), is used as the upper boundary condition (IBS$_{L12}$). This case represents the ideal scenario to generate the pseudo-reconstructed GSTs. On the other hand, the 2 m air temperature model output is employed (IBS$_{SAT}$) in order to obtain an ideal case of pseudo-reconstructed SATs. We use the ST$_{L12}$ as the reference GST to force the IBS$_{L12}$ forward model because in the upper 10 layers, the CLM4 is hydrologically active and it is interesting to keep this realism in the synthetic BTPs. Additionally, this depth is consistent with using annual resolution in the model data as at this level the annual wave is very damped in amplitude and layers below start to filter out decadal timescales. Nevertheless, this decision does not influence results significantly.





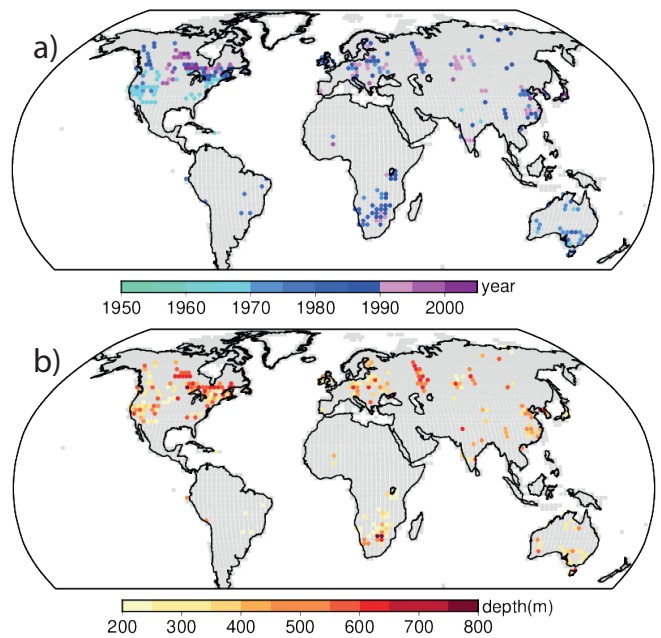

**Figure 1.** Spatial distribution of a) the logging date and b) the depths in the actual borehole network.

Second, a more realistic arrangement of the available global borehole network is implemented ($B_{mask}$), including the actual
spatial distribution of the borehole sites as well as their real logging depths and dates. This information is obtained from
the "Global Database of Borehole Temperatures and Climate Reconstructions" (Huang and Pollack, 1998). Only the BTPs
deeper than 200 m are considered in this step since the focus is on multi-decadal to centennial timescales within the LM
as in previous reconstructions works (e.g. Huang et al., 2000; Beltrami and Bourlon, 2004). After this selection, a total of

970 logs are retained. For this set up, each of the borehole sites is placed on a specific land model grid point according to
their geographic information (Fig. 1). If more than one borehole corresponds to the same model grid point, only one of them is
retained prioritizing the most recent one to allow for the climatic signal of the last decades of the 20th century to be represented.
Additionally, for the bulk of the cases, the most recent borehole records coincide with the deepest ones. We finally kept 301 grid
points that are nearest co-located to any of the real borehole locations. Once the spatial distribution of the borehole network

is represented in the CESM-LME grid, the LM $ST_{L12}$ series at each of these grid points is trimmed at the actual logging date
according to the date distribution (Fig. 1a). Then, the temperature anomalies are calculated with respect to the trimmed period
mean. The resulting anomaly time series are subsequently used to force the forward model at each grid point, generating a
pseudo BTP that is shortened to the actual borehole depth (Fig. 1b), in order to keep this configuration as realistic as possible.
In this configuration, the inversion of the individual profiles yields information until the date when they were logged, therefore

the reconstructed period is highly variable. The latter seemingly has an impact on the estimation of global averages since closer




to present day the number of available sites decreases. Indeed ca. half of the borehole profiles were logged before 1980 CE, and only ~20% of the sites have been measured after 1990 CE.

The results from the idealized IBS and the more realistic $B_{mask}$ configuration are then compared in order to evaluate the methodological limitations as well as the potential physical bias. As the methodological aspects are related to the suitability
of the spatio-temporal distribution of the borehole network to retrieve the past GST variations, the $IBS_{L12}$ allows for assessing these type of limitations if compared to the $B_{mask}$; the differences between them informing about the effect of the methodological variants. This assessment is initially developed at global scale, and then, it is extended to smaller spatial domains in order to address the implications on areas with different spatio-temporal borehole distribution. In this part of the analysis only the ALL-*F* ensemble is used.

The physical-related aspects and their influence on the estimates are subsequently explored via the comparison with the $IBS_{SAT}$, because any deviation from the pseudo-reconstructed SAT would be the result of the physical SAT-GST decoupling. Moreover, while the differences between the $IBS_{SAT}$ and the $IBS_{L12}$ cases would inform about the individual contribution of the physical aspects on the interpretation of the SAT variations from the pseudo-reconstructed GST, and the differences bewteen the $IBS_{L12}$ and $B_{mask}$ include the effect of the methodological sampling issues described above, the differences between $IBS_{SAT}$ and
the $B_{mask}$ allow for the estimation of the combined effects of both the methodological constraints and the physical processes. Besides the ALL-*F* ensemble, in this analysis the GHG- and LULC-only ensembles are also considered in order to estimate their specific contribution to the potential physical biases. Therefore, the previous PPE setups ($IBS_{SAT}$, $IBS_{L12}$, $B_{mask}$) are run for each of the simulations in the ALL-*F*, GHG- and LULC-only ensembles.

## 4   Results

The results of generating a variety of PPEs are reported herein. First, those related to methodological sampling issues (Sect. 4.1) and second, addressing SAT-GST coupling (Sect. 4.2).

### 4.1   Methodological issues

The comparison of the simulated global GST anomalies within the $IBS_{L12}$ and the $B_{mask}$ pseudo-reconstructions in the ALL-$F_2$ and ALL-$F_5$ members of the ALL-*F* ensemble are represented as an example in Fig. 2a. These members were selected
because they represent two possible results of the pseudo-reconstructed GST in the $B_{mask}$ configuration that are discussed herein. Similar results are obtained if other members are selected. In general, the $IBS_{L12}$ pseudo-reconstruction reasonably reproduces the gross features of the low-frequency GST variations over the LM in the CESM-LME. For instance, the transition from the MCA to a colder LIA and the warming over industrial times are successfully captured in both cases. Within the LIA and back to the MCA, the filtering effect produced by the heat diffusion averages over intervals of multidecadal and centennial
warming and cooling. Changes in the past, like the simulated MCA warming get damped in BTPs because of this effect and a very smooth version of them is recovered by the reconstruction. Nevertheless, in model experiments that simulate larger MCA-





LIA changes the borehole reconstruction is able to recover somewhat warmer temperatures during the MCA (González-Rouco et al., 2006, 2009).

Therefore, the $IBS_{L12}$ reproduces qualitatively the long term trends in both examples of Fig. 2a. However, the masked

inversion ($B_{mask}$) reproduces identical results in the case of ALL-$F_5$ but diverges from ALL-$F_2$ in the last decades of the 20th century. This would suggest that the results of $B_{mask}$ can be simulation dependent, i.e. dependent on the different initial conditions and therefore on internal variability. In order to evaluate this, a more quantitative estimation of trends is needed that allow for comparing the idealized and masked pseudo-reconstructions and also both of them with the simulated time series. To facilitate this, two approaches are proposed. One of them is to make a simple linear fit of the temperatures, either simulated

or pseudo-reconstructed over a reference period considered. As this approach can potentially be affected by the different discretization of the simulated (annual resolution) and pseudo-reconstructed (15-yr time steps), a second strategy is applied in which GST series are transformed to 15-yr averages coinciding with the time steps of the pseudo-reconstructed inversions over the time interval considered. Trends are then calculated by subtracting the mean values of the last and first 15-yr steps and dividing it by the total length of the reference time interval. This allows for verifying the robustness of results. We focus

on the 20th century to evaluate trends because: the bulk of the warming takes place in this period; results are less sensitive to the selection of the 15-yr interval than if the 19th century is considered, due to the influence of natural (internal and forced) variability; and it offers the possibility to compare results with instrumental trends (Hartmann et al., 2013). Thus, Fig. 2b shows results for linear fit to the 1900-2005 CE period and for trends calculated from the rates of change between 1890-1905 and 1990-2005 CE (15-yr-diff in Fig. 2b). Figure 2b shows the frequency distribution of trends calculated from both approaches

for all members within the ALL-$F$ ensemble. Box-and-whisker plots are shown for all possible scenarios considered herein: GST, $IBS_{L12}$, GST masked with the realistic borehole configuration ($GST_{mask}$), $B_{mask}$ as well as the differences among them ($IBS_{L12}$ - GST, $B_{mask}$ - $GST_{mask}$, GST - $GST_{mask}$ and $IBS_{L12}$ - $B_{mask}$).

Interestingly, the frequency distribution of trends within the ALL-$F$ ensemble is similar for both strategies (15-yr-diff and linear fit). The estimated global trends (GST in Fig. 2b) range between 0.3 and 0.6 K century$^{-1}$ across the 13 simulations. Thus,

internal variability has an impact in these trends estimates. The values are somewhat smaller than those of the observational record (Hartmann et al., 2013). In both cases, the $IBS_{L12}$ pseudo-reconstruction yields a reasonable estimation of the global GST increase during the 20th century. Note that the GST trends are slightly larger for the 15-yr-diff relative to the linear fit. While the linear trend indicates a median GST increase of 0.39, the 15-yr-diff suggests a median of 0.47 K century$^{-1}$. Nevertheless, the $IBS_{L12}$ - GST differences are distributed around zero in either case (Fig. 2b). The comparison of the two

methods to estimate the 20th century temperature increase shows that in either case, the pseudo-reconstructed GSTs robustly represent the targeted temperature signal. It is remarkable that when GSTs are masked, i.e. sample, in space, depth and time ($GST_{mask}$) following the real distribution, trends take a smaller range of values than those of GST. GST - $GST_{mask}$ differences are above zero and can reach 0.3 K century$^{-1}$, thus significant and important in the context of the simulated warming. It is remarkable how the level of impact may depend on the interplay of internal variability and BTP sampling. Additionally, the

pseudo-reconstructions $B_{mask}$ deliver correspondingly a distribution of trends that agrees in range with those of $GST_{mask}$. In the





**Figure 2.** a) LM global GST annual anomalies and the corresponding 31-yr filtered output, the global $IBS_{L12}$ and the $B_{mask}$ pseudo-reconstructions for the ALL-$F_2$ and ALL-$F_5$ members of the ALL-$F$ ensemble. Note the different discretization in the x axis after 1700 CE. b) Boxplots describing the centennial trends over the period 1900-2005 CE calculated for each of the 13 ensemble members within the ALL-$F$ after applying space, depth and time masking in the model to mimic real BTP distribution. c) as in b) but applying spatial and depth masking (right) and spatial only masking (left). The GST, $IBS_{L12}$, $GST_{mask}$, $B_{mask}$ and the differences between $IBS_{L12}$ - GST, $B_{mask}$ - $GST_{mask}$, GST - $GST_{mask}$ and $IBS_{L12}$ - $B_{mask}$ cases are represented. Trends are presented as the 15-yr-diff (b left; see text) and the linear fit to the data calculated over an annual basis (b right). The 25th, 50th and 75th percentiles are indicated and the whiskers represent the lowest/highest value within 1.5 interquartile range (IQR) of the 25th/75th percentile. Outliers are indicated as diamonds in the same color of the box. They represent the values lower/higher than the lower/upper whisker. Note that colors in a) correspond with those in b) and c)





case of 15-yr-diff $B_{mask}$ underestimates slightly, with $B_{mask}$ - $GST_{mask}$ distributing slightly below zero, whereas in the linear case $B_{mask}$ - $GST_{mask}$ are centered over zero.

Therefore, the borehole reconstructions are able to retrieve the masked or unmasked GST. However, sampling plays a role and can produce an underestimation of GSTs. This underestimations depends on the interplay between sampling and internal variability and occurs in some simulations in which the selected points their depths and logging dates are not optimal to represent global warming. Note that in Fig. 2b $B_{mask}$ actually includes the effects of masking, i.e. selecting, pseudo-BTPs co-located to the actual BTP network, trimmed to their actual depths in reality and generated using GST histories up to their logging dates. In turn, $GST_{mask}$ includes the effects of masking, i.e. selecting in this case grid-point time series co-located to actual BTP locations and trimmed to their logging dates; depth masking does not play a role in the case of GST series.

At this point, the question remains on what is the relative role of each of the three masking effects. Figure 2c addresses this by showing similar plots as the linear fit in Fig. 2b but considering only spatial and spatial plus depth masking. The results for GST, $IBS_{L12}$, their masked versions and differences are identically shown as in Fig. 2b (right) for comparison. Results are virtually identical for both plots in Fig. 2c. This implies that the effects of depth masking are negligible. Spatial masking does have an impact, albeit smaller than if the effects of logging dates are considered, with GST - $GST_{mask}$ differences above zero and below 0.2 K century$^{-1}$. $IBS_{L12}$ and $B_{mask}$ are effective in retaining their targets.

The selection of the number of singular values to be retained in the SVD inversion may also exert some influence in the estimation of the GST recent trends. Therefore, the impact on reconstructed GST trends of including four and five PCs has been analyzed. The latter is illustrated in Fig. S1 of the supplementary material. On one hand, the solution considering the four leading modes yields similar estimations than the solution based on the three leading modes. Nonetheless, the 4 PCs $IBS_{L12}$ solution is slightly biased to larger values (Fig. S1a). Such behavior is also present in the $B_{mask}$ configuration relative to the $GST_{mask}$. This pattern is systematically observed in all members of the ALL-*F* ensemble as indicated in the Box-and-whisker plot in Fig. S1a. Note the positively biased $IBS_{L12}$ - GST and $B_{mask}$ - $GST_{mask}$. On the other hand, the results including the five leading modes yield a solution with some increase of the variance (Beltrami and Mareschal, 1995), noticeable within the 20th century. The simulated GST 20th century trends are biased to smaller values in comparison to those in Fig. 2b; systematically underestimated by the $IBS_{L12}$ 5 PCs solution (Fig. S1b).

The hemispheric to global borehole reconstructions in the real-world conditions are far from the idealized scenario described by the $IBS_{L12}$ since they are the result of an aggregation of a limited amount of BTPs that are sparsely distributed over the land surface. Further, there is a large variability in both depth and timing of the records since BTPs are obtained from different sources and they are rarely drilled for the development of climate studies (Jaume-Santero et al., 2016).

The global GST increase during industrial times is however underestimated by the $B_{mask}$ scenario. This feature is common to all ensemble members within ALL-*F* pool of simulations, being the median $IBS_{L12}$ - $B_{mask}$ difference of 0.17 K century$^{-1}$ (Fig. 2b), which accounts for about 43% of the simulated global GST 20th century warming. Most of this underestimation is due to temporal masking. Almost half of the grid points containing BTPs in the $B_{mask}$ case are dated prior to 1982 CE and only ~5% of them present logging dates after 1995 CE (Fig. 1a). As a consequence, many of the synthetic BTPs do not include the information of the last two decades of the simulated period leading to a muted estimation of the global GST recent trend



(Pollack and Smerdon, 2004). The variability of the depth of the borehole records, on the other hand, has no influence in the results of the $B_{mask}$ configuration. This is because only BTPs deeper than 200 m have been retained, a depth that has been shown to be sufficient to capture the trend estimates from the LIA to present days and to even retain some features of the MCA to the LIA transition. Likewise, the spatial distribution of the borehole records has a smaller influence on the $B_{mask}$ trends at

hemispheric to global scales as indicated in both real-world borehole reconstructions (Pollack and Smerdon, 2004; Beltrami and Bourlon, 2004) and PPEs (González-Rouco et al., 2006).

In spite of the general pattern of discrepancies between $B_{mask}$ and $IBS_{L12}$ during the industrial period (e.g. ALL-$F_2$; Fig. 2a), there are also cases in which both estimates show a good agreement (e.g. ALL-$F_5$). This range of variability in the representation of the 20th century trends indicates that the internal climate variability within each realization exerts some

influence in the simulated trends of the $B_{mask}$ configuration. As the global average for the last two decades highly depends on the most recently logged BTPs, mostly concentrated in northern Canada, central Europe, Russia and Japan (Fig. 1a), the internal climatic response over these regions apparently play a significant role on the global 20th-century trend estimations. Nevertheless, it would be desirable to expand and update the current dataset of BTPs. In the meantime, it seems advisable to derive strategies that minimize the impact of aging in BTPs (Harris and Chapman, 2001), perhaps by blending BTP information

and instrumental temperature observations in the last decades or other procedures that reduce underestimation of multidecadal trends during this period.

The variability in the spatiotemporal distribution of the borehole network may impact the representation of the past GST evolution from BTPs at smaller spatial scales. In order to explore this, three sub-continental domains with different levels of BTPs coverage have been selected. These domains include regions over North America, Europe and Africa. Their selection

intents to be illustrative of sampling effects by broadly including existing BTPs in each domain although without necessarily representing optimized domain configurations in any sense. The GST annual anomalies for these regions as well as their corresponding 31-yr running mean low-pass filter outputs, the $IBS_{L12}$ and the $B_{mask}$ estimates cases are shown for one member of the ALL-$F$ ensemble as an example for each of the regions (Fig. 3 center). The ALL-$F_2$ is presented for North America (Fig. 3a) while the ALL-$F_3$ is shown for both Europe (Fig. 3b) and Africa (Fig. 3c) as these members are representative of the

mean behavior of the $B_{mask}$ configuration within the ALL-$F$ ensemble. The Box-and-whisker plots (Fig. 3 right) show the 20th century trends from the 13 members of the ensemble as in Fig. 2b using the linear regression. Interestingly, the results of the $IBS_{L12}$ and the $B_{mask}$ configurations show better agreement over the better sampled area of North America than over Europe and Africa. This is evident in the representation of the 20th century trends with a difference in their corresponding median value around 0.08 K century$^{-1}$ in North America. Masking produces differences (GST - GST$_{mask}$, $IBS_{L12}$ - $B_{mask}$ in Fig. 3) that

range from zero to 0.2 K century$^{-1}$, thus suggesting underestimation biases. The bias is reduced when only spatial masking is considered (Fig. 4), although some outliers still produce GST - GST$_{mask}$ differences above 0.1 K century$^{-1}$. The size of those differences is not large but it represents between 20-30 % of the simulated trends.

The results for Europe and Africa are more variable. GST trends center around 0.4 K century$^{-1}$ in North America and Europe and below 0.1 K century$^{-1}$ in the African domain considered (Fig. 3). However, trends at these spatial scales can be highly

dependent on internal variability and some simulations show no significant trends over the European and African domains.





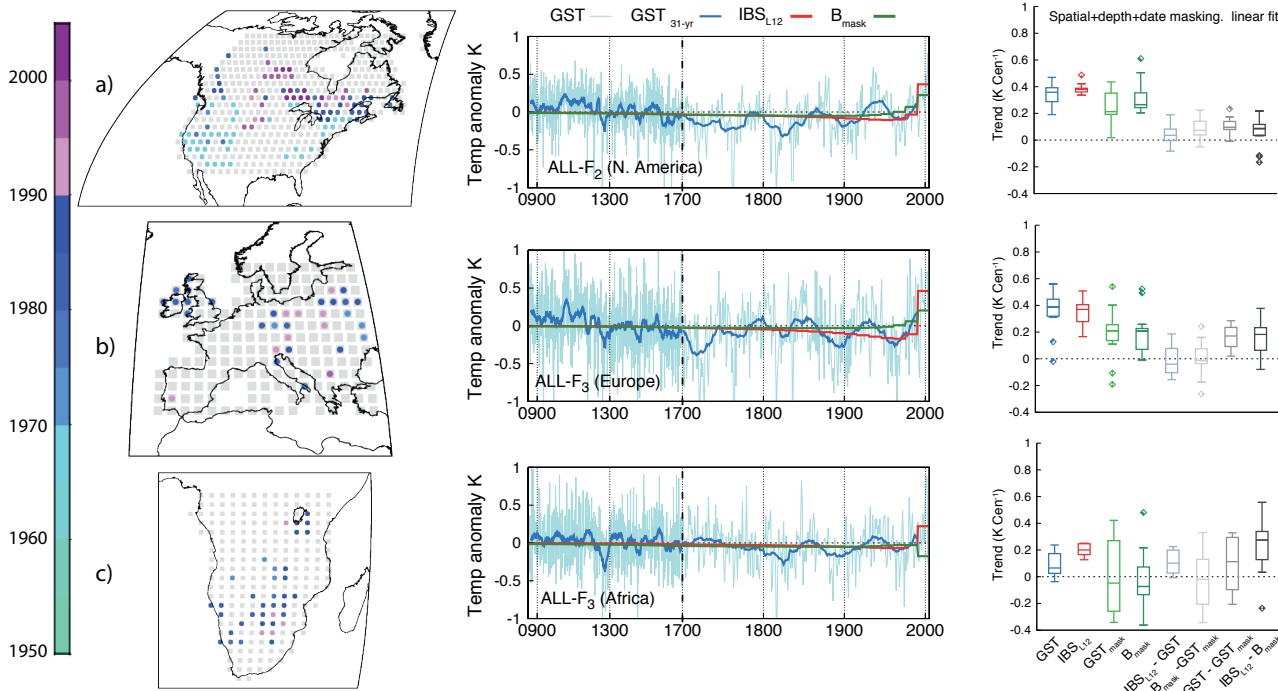

**Figure 3.** LM global GST annual anomalies and pseudo-reconstructions for three different sub-continental regions: a) North America (31.26-69.16° N, 135-55° W), b) Europe (36.59-59.68° N, 10° W - 30° E) and c) Africa (35.05 ° S - 2.28° N, 10-40° E). The ALL-$F_2$ member is displayed for the North American continent, whereas the ALL-$F_3$ member of the ALL-F ensemble is used for the European and African regions. Maps on the left show the spatial distribution of BTPs locations and dates of the actual borehole network for each of the regions. Grey dots show the model grid and the areas defining each of the regions. Right panels: as in Fig. 2a,b but with the 1900-2005 CE trends presented only for the linear fit method.

Additionally, poor spatial sampling enhances the influence of local behavior. Note that the representation of the 20th century trends in both the $GST_{mask}$ and the $B_{mask}$ spreads over a larger range than for North America, especially for the African region. This happens as a response to a decline in the availability of recent BTPs to calculate the regional averages during the last decades of the simulated period. In Europe, the $B_{mask}$ configuration systematically underestimates the $IBS_{L12}$ results, but there

are cases (not shown) in which the results of the $B_{mask}$ case closely matches the ideal scenario. Thus, these differences among estimates from the different ensemble members suggest an influence of the internal variability on the estimations of the recent temperature trends. Over the African continent, the estimates are more diverse. Therein, the difference between the $IBS_{L12}$ and $B_{mask}$ depicts a larger variability, with a general poor representation of the 20th century GST increase over this region.

When only spatial masking is considered the spread of results is reduced for all regions. For the European domain most of

the solutions of GST - $GST_{mask}$ ($IBS_{L12}$ - $B_{mask}$) get confined below +0.1 (-0.1) K century$^{-1}$ (Fig. 4). For the African domain, the dispersion in GST - $GST_{mask}$ and $IBS_{L12}$ - $B_{mask}$ also shrinks, with both cases indicating overestimation than can be larger





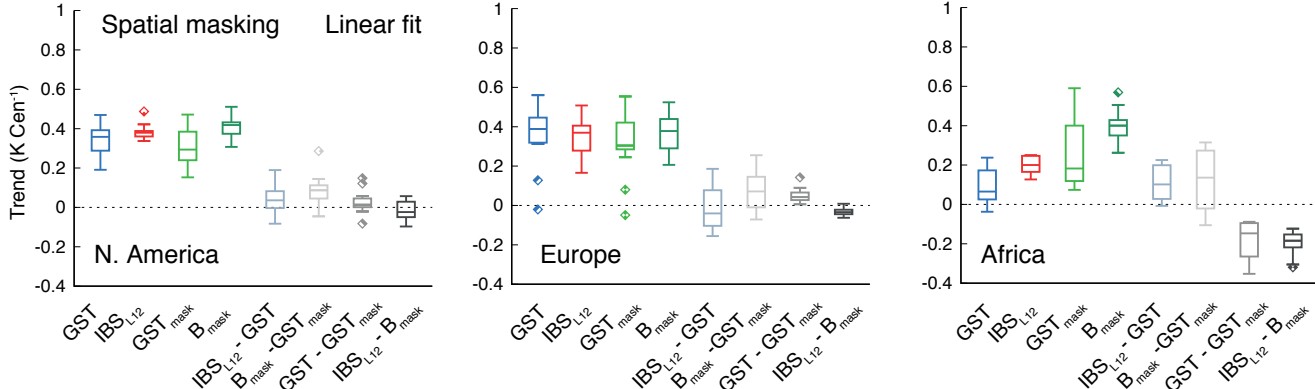

**Figure 4.** As in Fig. 3 (right) but considering only spatial masking. Linear fit estimated 1900-2005 CE trends in the ALL-*F* ensemble for North America, European and African regions shown in Fig. 3. Acronyms for GST and PPE cases and their differences follow the same convention as in Figs. 2 and 3.

than 0.3 K century$^{-1}$ when spatial masking only (Fig. 4) is considered, i.e. the selected grid points indicate larger warming than the rest of the regions. The selected case examples in Fig. 3 (center) illustrate how different the unmasked IBS$_{L12}$ and the B$_{mask}$ solutions can be both in representing the warming of the last decades of the 20th century and also de cooling during the 19th

and early 20th century. The latter is most noticeable in the example shown for Europe where clear differences develop during the 19th century between IBS$_{L12}$ and B$_{mask}$. Here, spatial sampling misses the cooling shown by IBS$_{L12}$. The reasons for that are discussed in Sect. 4.2.

Therefore, the B$_{mask}$ 20th century trends are sensitive to the spatiotemporal distribution of the borehole network particularly at smaller scales. Whereas in the North American continent there is a better coverage of borehole logs, including most of

those recently recorded, in Europe, and especially in Africa, there is a comparatively poorer representation and inhomogeneous spatial distribution of BTPs and particularly, of the recent ones, that ultimately impact the resulting temperature trend estimates. This suggests that the interpretation of the trends from borehole reconstruction estimations at the regional scale should be done with caution over areas with poor spatiotemporal coverage of BTPs (Huang et al., 2000).

## 4.2 Biases of past SAT borehole-based reconstructions due to physical processes

In addition to the limitations imposed by the spatio-temporal borehole distribution, the reconstructed SAT can also be biased by changes in the long-term SAT-GST relationship. Specifically, changes in anthropogenic forcing after 1850 CE can contribute to SAT-GST decoupling (Melo-Aguilar et al., 2018). Figure 5a-c shows the LM evolution of SAT and GST anomalies relative to the 850-2005 CE mean as well as the difference between both and the linear trend of SAT-GST for the period 1850-2005 CE at selected grid points, illustrating three possible behaviors of the long-term SAT-GST relationship in the CESM-LME. First,

a case in which this relationship remains stable during the whole LM (Fig. 5a). Note that there is no trend in the SAT-GST



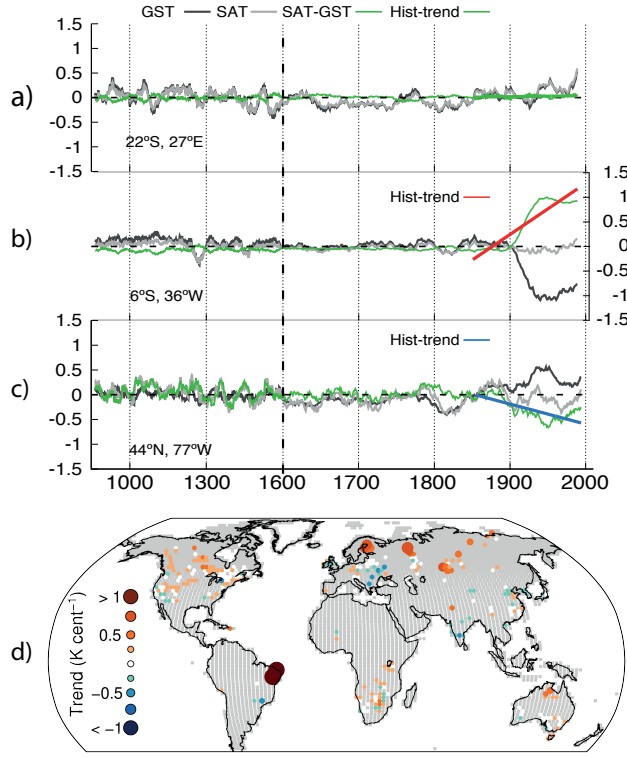

**Figure 5.** LM evolution of SAT and GST anomalies, SAT minus GST and the linear trends of SAT minus GST over the 1850-2005 CE period for three different grid points with borehole records in the ALL-$F_2$ as an example. Each grid point illustrates a possible case of the long-term SAT-GST relationship: a) strong coupling during the LM, b) decreasing of GST relative to SAT and c) increasing of GST relative to SAT during industrial times. d) Spatial distribution of linear trends in SAT minus GST anomalies over the 1850-2005 CE period evaluated at every grid point with the presence of a borehole site using the ALL-$F$ ensemble mean. Grid points showing statistically significant trends ($p < 0.05$) are colored in red/blue depending on whether they are increasing/decreasing SAT-GST trends.

differences. This case represents the ideal strong SAT-GST coupling situation from which the GST would constitute a good proxy of the SAT. Second, two different cases in which the SAT-GST long-term relationship experiences significant variations are depicted. The direction and magnitude of the SAT minus GST trend are suggestive of the type of impact on the SAT-GST coupling. In the first case (Fig. 5b), the sharp decrease of GST around 1900 CE results in a positive trend in the SAT-GST differences of 1.04 K century⁻¹, that is represented in red indicating a warmer SAT relative to the GST. The second one (Fig. 5c), depicts an example in which the SAT tends to be colder than GST during the industrial period thus, leading to a negative trend of about -0.4 K century⁻¹ (blue). These two cases are suggestive of a physical interference affecting the temperature signal contained in the subsurface. Therefore, the inversion of BTPs under such characteristics would yield unreliable information of





the past SAT variations. The ALL-$F_2$ simulation is used as example but similar results can be found in other ALL-$F$ ensemble
members.

To provide a spatial view of the borehole locations affected by changes in the long-term SAT-GST relationship in the CESM-
LME world, we evaluate the SAT minus GST industrial (1850-2005 CE) trends for the ALL-$F$ ensemble mean at every grid
point with the presence of a borehole (Fig. 5d). The ensemble mean is used in order to identify the forced response, however,
the internal climate variability in individual ensemble members may result in a different representation of both the magnitude
and sign of the trends (Melo-Aguilar et al., 2018). Since the industrial period is affected by pronounced temperature trends
due to the anthropogenic emissions of GHGs, the 1850-2005 CE interval is the most adecuate to evaluate the particularities
of the SAT-GST linear trends. Additionally, the analysis in this part is intended to determine the reliability of the borehole
technique to retrieve the SAT increase over the industrial period. Figure 5d illustrates that at a large number of grid points
containing BTPs, the SAT minus GST linear trends are statistically significant ($p < 0.05$) identifying therefore some level of
SAT-GST decoupling during the industrial period at such locations. These grid points are represented in red (blue) if the trend
of the temperature differences is positive (negative). Additionally, the size of the circles is related to the magnitude of the trend.
Therefore, both color and size indicate the direction and magnitude of the SAT-GST decoupling. Note that positive trends are
dominant with some larger values at north-eastern Brazil, Fennoscandia, central Eurasia and the north of Siberia. Negative
values are also evident, mostly distributed around central and eastern Europe and some more isolated cases distributed around
the globe.

In order to assess the influence of the detected long-term SAT-GST decoupling on the representation of SAT from the
borehole-based reconstructions at global scale, results of the SAT and GST trends and their IBS pseudo-reconstructions, as
well as the $B_{mask}$ case including the effects of sampling are shown in Fig. 6, together with the ALL-$F_2$ and ALL-$F_5$ ensemble
members as an example. In both of them, the simulated SAT low-frequency variations over the LM are broadly reproduced
by the IBS$_{SAT}$. Note that the simulated SAT 20th century trend is accurately captured by the IBS$_{SAT}$ since the differences
between ensemble member trends are distributed around zero (Fig. 6b). Therefore, the IBS$_{SAT}$ can be considered a reasonable
representation of the simulated SAT, specifically in the estimation of the 20th century warming. Thus, the comparison between
IBS$_{SAT}$ and IBS$_{L12}$ will be informative of any deviation between the pseudo-reconstructed GST and the simulated SAT. Indeed,
SAT-GST trend differences distributed over a median value of 0.1 K century$^{-1}$ and consistently, the IBS$_{SAT}$ - IBS$_{L12}$ 20th
century trends differences yield positive values, with a median of about 0.11 K century$^{-1}$ and a high level of agreement, i.e.
small dispersion, among the members of the ALL-$F$ ensemble (Fig. 6b). This indicates that even under an ideal scenario, the
pseudo-reconstructed GST does not fully capture the 20th century SAT warming, missing on average ~20% of the simulated
SAT increase in the CESM-LME as a response to the long-term SAT-GST decoupling. Furthermore, the effect of the physical
processes is superimposed to the limitations due to the methodological aspects, leading to larger differences with respect to the
more realistic pseudo-reconstructed GST ($B_{mask}$). In fact, the IBS$_{SAT}$ - $B_{mask}$ 20th century trends differences have a median of
0.28 K century$^{-1}$ (Fig. 6b) that represents more than 50% of the simulated SAT. There is however a larger variability of trend
differences between these two scenarios that ranges between 0.16 and 0.43 K century$^{-1}$. This spread results from the internal





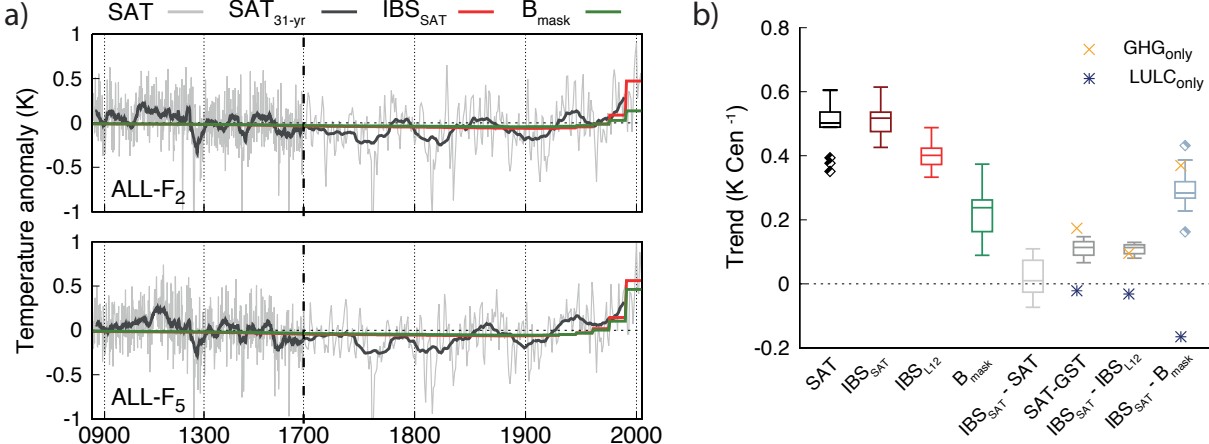

**Figure 6.** a) LM global SAT annual anomalies and the corresponding 31-yr filtered outputs, the global $IBS_{SAT}$ and the $B_{mask}$ pseudo-reconstructions for the $ALL-F_2$ and $ALL-F_5$ members of the $ALL-F$ ensemble. Note the different discretization in the x axis after 1700 CE. b) Estimated linear fit as Fig. 2b, but the SAT, $IBS_{SAT}$, $IBS_{L12}$, $B_{mask}$ and the differences between $IBS_{SAT}$ - SAT, SAT - GST, $IBS_{SAT}$ - $IBS_{L12}$ and $IBS_{SAT}$ - $B_{mask}$ cases are represented. Additionally, the ensemble mean of the GHG- and LULC-only ensembles is indicated by the crosses and asterisks, respectively, in the $IBS_{SAT}$ - $IBS_{L12}$ and the $IBS_{SAT}$ - $B_{mask}$ columns.

climate variability in each realization of the ensemble. Such range of variability is noticeable in Fig. 6a if $IBS_{SAT}$ and $B_{mask}$ trends for the two ensemble members are compared to each other.

Melo-Aguilar et al. (2018) showed that the long-term SAT-GST decoupling in the CESM-LME is mainly driven by LULC and GHG changes. LULC changes modify the radiative fluxes at the the surface leading to a different response of the SAT and GST. Likewise, the SAT increase during industrial times as a response to the increase in GHGs may not be fully transferred to the soil due to the insulating effect of snow cover feedbacks. To explore their effects on borehole reconstructions, the analysis described in Fig. 5d is extended to the GHG- and LULC-only ensembles (Fig. 7a). The GHG-only ensemble (Fig. 7a left) the

dominant effect is represented by positive SAT minus GST trends, with a similar pattern to the $ALL-F$ ensemble over North America, Fennoscandia, northern Russia and Siberia in Fig. 5, although the magnitude of trends is significantly larger. On the contrary, in the LULC-only ensemble negative SAT minus GST estimates are evident, indicating a similar response as in the $ALL-F$ ensemble over central and eastern Europe and the Indian subcontinent (Fig. 5). Additionally, the strong positive trends over north-eastern Brazil resemble those found in the $ALL-F$ ensemble.

The $IBS_{SAT}$, the $IBS_{L12}$ and the $B_{mask}$ configurations are also implemented in the GHG- and LULC-only ensembles in order to evaluate their contribution to the physical SAT-GST decoupling at global scale. Figure 7b compares the simulated global SAT anomalies with the $IBS_{SAT}$ and the $B_{mask}$ pseudo-reconstruction in the $GHG_1$ (Fig. 7b left) and $LULC_1$ (Fig. 7b right) members of the GHG- and LULC-only ensembles, respectively, as an example. From a simple visual inspection it may be noticed that the $GHG_1$ simulation portrays a similar response than the $ALL-F$ estimates (Fig. 6a) since the $IBS_{SAT}$ and the



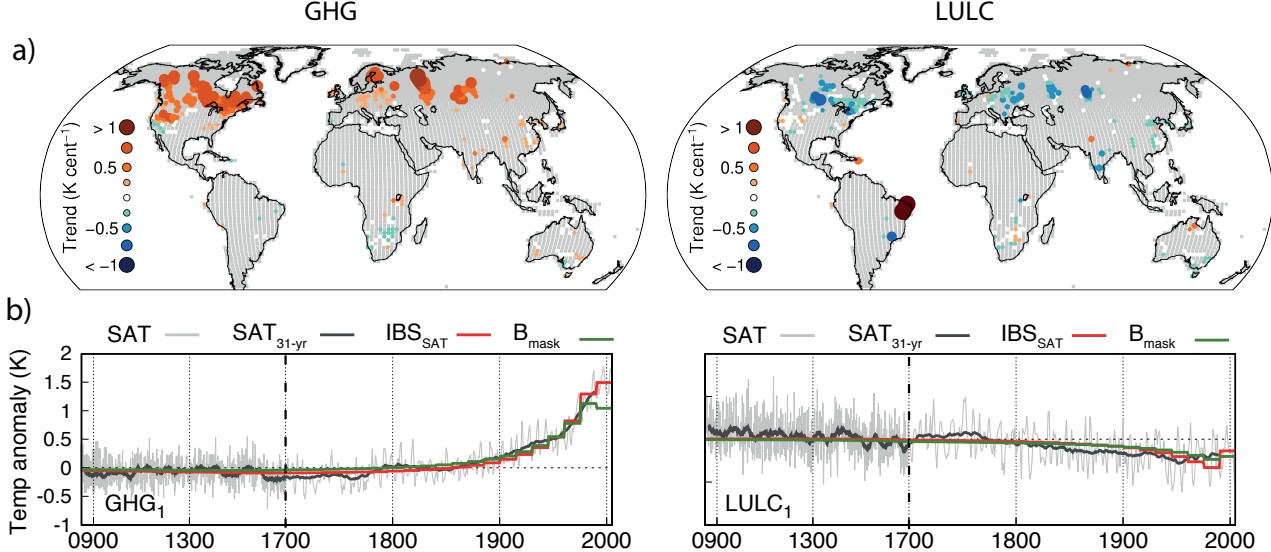

**Figure 7.** a) Spatial distribution of linear trends in SAT - GST anomalies over the 1850-2005 CE period evaluated at every grid point co-located to a borehole site in the GHG- and LULC-only ensemble mean (left and right, respectively). Only grid points delivering statistically significant trends ($p < 0.05$) are colored. b) LM global SAT annual anomalies and the corresponding 31-yr filtered outputs, the global IBS$_{SAT}$ and the B$_{mask}$ pseudo-reconstructions for the GHG$_1$ and LULC$_1$ ensemble members. Note the different discretization in the x axis after 1700 CE.

B$_{mask}$ cases diverge during the last decades of the simulated period. On the other hand, in the LULC$_1$ case (Fig. 7b right), there is not such similarity to the ALL-*F* cases. All series and pseudo-reconstructions suggest a consistent cooling in response to LULC changes throughout the last centuries of the millennium. This analysis suggests a larger contribution of the GHG forcing to the SAT-GST decoupling relative to the LULC forcing at the global scale.

To provide a quantitative support to this statement the GHG- and LULC-only ensemble mean differences between the IBS$_{SAT}$
- IBS$_{L12}$ and IBS$_{SAT}$ - B$_{mask}$ 20th century trends are included in Fig 6b (crosses and asterisks, respectively). Note there that in the IBS$_{SAT}$ - IBS$_{L12}$ case the GHG-only ensemble shows positive and very similar estimates than the median of the ALL-*F* ensemble (0.09 and 0.11 K century$^{-1}$, respectively), suggesting a large contribution of the GHG forcing to the physical bias in the representation of SAT by the borehole pseudo-reconstructions in the CESM-LME at global scale. On the contrary, in the LULC-only ensemble the IBS$_{SAT}$ - IBS$_{L12}$ difference is negative and very small (-0.03 K Century$^{-1}$) indicating a negligible
contribution at this spatial scale. These results are in agreement, as expected, with pure SAT-GST differences, that also receive a larger contribution from GHG at these spatial scales. The IBS$_{SAT}$ - B$_{mask}$ differences further highlight the larger influence of the GHG relative to the LULC forcing. Whereas in the GHG-only ensemble the mean IBS$_{SAT}$ - B$_{mask}$ difference shows values in the same direction as the ALL-*F* ensemble, in the LULC-only ensemble the difference goes in the opposite direction (negative IBS$_{SAT}$ - B$_{mask}$ trend differences). Melo-Aguilar et al. (2018) reported that the contribution from the GHG forcing





to the SAT-GST decoupling is controlled by the reduction in the NH winter snow cover as a response to higher temperatures
during industrial times. This situation increases the exposure of the soil surface, previously insulated by snow cover, to the cold
winter air, leading to an overall effect of warmer SAT relative to GST at a global scale.

Regionally, the influence of the SAT-GST long-term decoupling on the representation of simulated SAT from the pseudo-reconstructed GST deserves also to be considered since there are geographical variations of this effect (Fig. 5d). Figure 8
illustrates SAT annual anomalies with respect to the 850-2005 mean and the corresponding 31-yr low pass filter outputs as
well as the $IBS_{SAT}$ and the $B_{mask}$ cases for the ALL-$F_2$, $GHG_1$ and $LULC_1$ simulations over the same areas described in
Fig. 3. The spread provided by considering all members of each ensemble are depicted in the boxplots at the bottom of Fig.
8. Interestingly, the decoupling effect appears to be larger over North America and Africa than over Europe. As in the case
of the global average, the effect over North America and Africa is shown by the underestimation of the SAT warming during
industrial times, indicated by the positive SAT-GST median differences of ~0.2 K Century$^{-1}$ consistent with comparable ~0.2 K
Century$^{-1}$ median change in $IBS_{SAT}$ - $IBS_{L12}$ that endorse the performance of the borehole method. On the contrary, in Europe,
the SAT-GST decoupling leads to small differences, a negligible under- (over-) estimation of the SAT increase as shown by
SAT-GST ($IBS_{SAT}$ - $IBS_{L12}$).

The single-$F$ experiments indicate variability in the influence of both the GHG and LULC forcings over the different areas
considered herein. In North America there is a strong contribution of the GHG forcing, that is somewhat counteracted by the
LULC influence. This is noticeable not only in the time series in Fig. 8a but also in the sign and magnitude of the SAT-GST
and the $IBS_{SAT}$ - $IBS_{L12}$ mean difference of both the GHG- and LULC-only ensembles (Fig. 8a bottom). Conversely, over the
European region, the largest contribution comes from the LULC forcing (see the larger differences in SAT-GST and $IBS_{SAT}$
- $IBS_{L12}$ for LULC-only in Fig. 8b bottom). This is shown more clearly in the comparison of the selected examples in Fig.
8b for ALL-$F_2$, $GHG_1$ and $LULC_1$, that show that the LULC cooling dominates the long term trends over GHG warming
in the ALL-$F$ experiment. For Africa, the ALL-$F$ experiment also evidences the damping of the GHG warming produced bu
LULC negative trends. However, other external forcings may also contribute to the response of the SAT-GST relationship at
the continental scale in this region (Melo-Aguilar et al., 2018). It is also remarkable that the masked reconstruction ($B_{mask}$) in
the ALL-$F$ and LULC cases in Fig. 8b does not capture the long term cooling during the 19th and 20th centuries shown by the
unmasked SAT averages (Fig. 8b) and GST averages (Fig. 3b center). Thus, the influence of LULC forcing also explains the
different trends between $IBS_{L12}$ and $B_{mask}$ in Fig. 3b (center) during the 19th and early 20th centuries.

The superposition of the physical biases and the methodological aspects at the continental scales suggest further comments.
For instance, for the North American region, while the methodological constraints have a relatively reduced impact on retriev-
ing the simulated GST 20th century increase in the $B_{mask}$ configuration (as discussed in Sect. 4.1), the effect of the physical
processes result in a larger underestimation of the SAT 20th century evolution. This is evident in the $IBS_{SAT}$ - $B_{mask}$ difference,
that has a median of 0.28 K century$^{-1}$ (Fig. 8), representing ~50% of the simulated SAT 20th century warming. The increment
can be compared with the $IBS_{L12}$ - $B_{mask}$ in Fig. 3right. The largest contribution comes from the GHG-only forcing, partially
reduced by the LULC-only effect, as shown by the single forcing crosses in Fig. 8. Similarly, in African region, the effects of
the physical processes contribute to enhance the underestimation of the SAT signal by the $B_{mask}$. $IBS_{SAT}$ - $B_{mask}$ differences

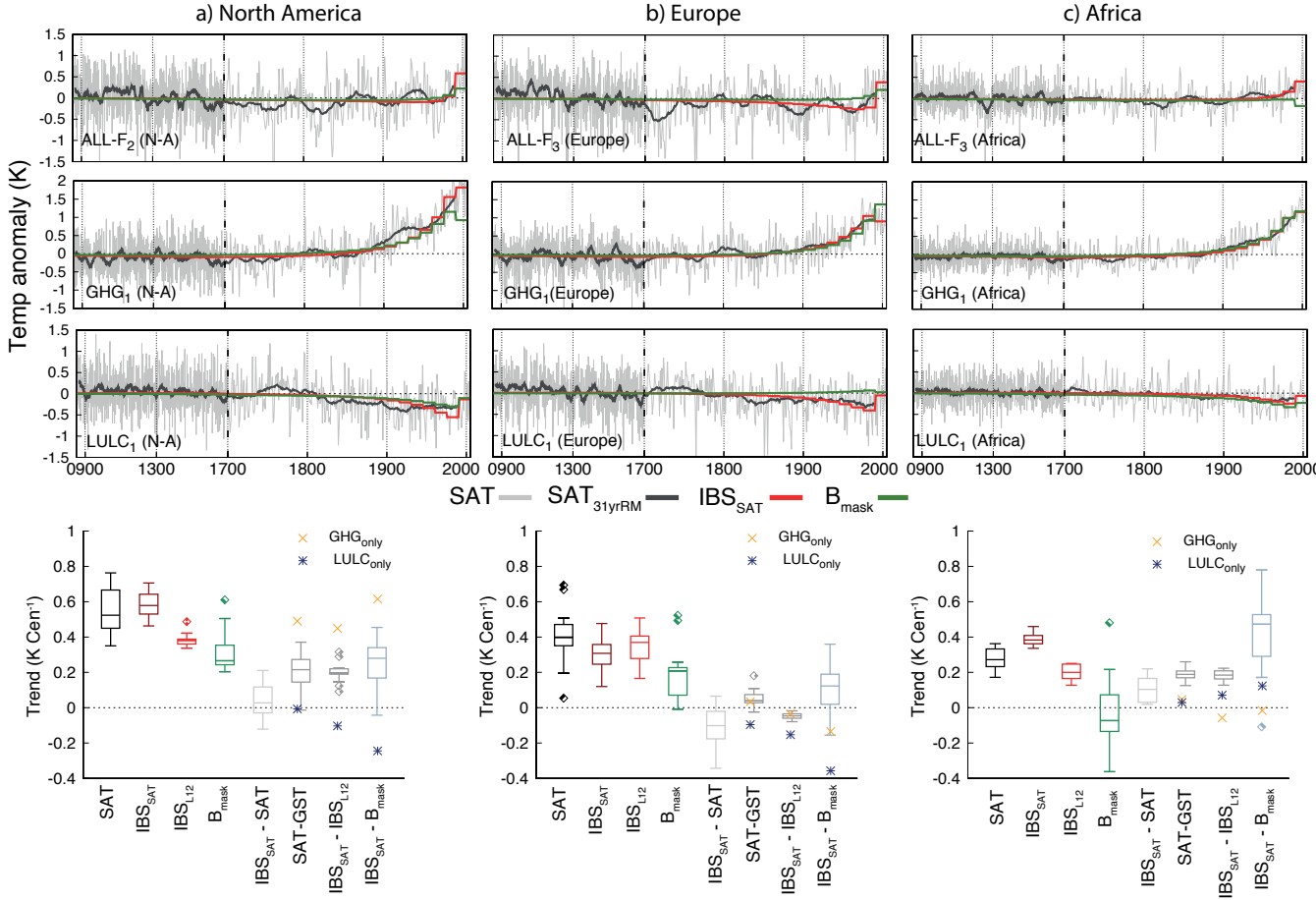

**Figure 8.** LM regional SAT annual anomalies and the corresponding 31-yr filtered outputs, the global $IBS_{SAT}$ and the $B_{mask}$ pseudo-reconstructions for the ALL-$F_2$, $GHG_1$ and $LULC_1$ (from top to bottom) members of the ALL-$F$, GHG and LULC-only ensembles, respectively: North America (a), Europe (b) and Africa (c). Note the different discretization in the x axis after 1700 CE. Bottom panels: as Fig. 6b but for each of the regions presented.

expand their spread in Fig. 3 reaching now values of 0.8 K Century$^{-1}$. On the contrary, for the European region the SAT-GST decoupling slightly counteracts the effects of the methodological aspects mostly via LULC negative biases. However, in both of these regions, the largest contribution to a biased estimation of the SAT 20th century trends comes from the methodological aspects (see SAT - GST and $IBS_{L12}$ - $B_{mask}$ differences in Fig. 3) that outweigh the bias from the physical processes.

## 5   Conclusions

Borehole based reconstructions lean on two hypothesis to derive the evolution of past temperature trends. One of them is that past GST histories can be recovered from BTPs where the conductive regime dominates; the second one is that the past



GST evolution is coupled to SAT changes and thus, the past history of SAT changes can be recovered from BTPs. The first hypothesis can be affected by methodological issues that may distort the recovery of past GSTs from BTPs, whereas the second hypothesis can be affected by physical issues that may distort SAT-GST coupling and thus, the recovery of past SAT changes.

This study analyses the performance of the borehole temperature reconstruction technique in a pseudo proxy framework that allows for addressing both methodological and physical issues.

Previous works using PPEs of borehole reconstructions have been implemented in simulations of the LM and focussed on the methodological performance at global (González-Rouco et al., 2006; García-García et al., 2016) and regional scales (González-Rouco et al., 2009), either using the output of a single climate model (González-Rouco et al., 2006, 2009) or an

ensemble of PMIP3/CMIP5 LM experiments (García-García et al., 2016). We have extended the analysis of previous works by implementing PPE strategies within the ensemble of simulations of the LM produced with the CESM climate model, the so-called CESM-LME (Otto-Bliesner et al., 2016). We have updated past analyses by introducing a more realistic PPE setup to address both methodological and physical issues at global and regional scales. Additionally, the use of an ensemble of simulations with the same model and different forcing boundary conditions has allowed for considering the influence of

internal variability and external forcings on the application of the borehole method.

The methodological implementation used herein adopts a standard SVD approach, as described in González-Rouco et al. (2006, 2009) and García-García et al. (2016). Similar to previous studies, the PPE has been developed in a so-called idealized scenario in which BTPs are assumed to exist at every land model grid point, and produced with a forward model using the complete simulated GSTs, 850-2005 CE. This is equivalent to consider that information from BTPs would be available everywhere

in land and with logging dates updated to present times. Thus, in addition, more realistic scenarios considering distributions of BTPs that mimic their actual spatial, depth and logging date distributions have been constructed. The former is used as a benchmark from which the performance of the realistic case is evaluated.

Regarding physical influences on the SAT-GST relationship, we build on from the results of Melo-Aguilar et al. (2018) that demonstrate with this ensemble of CESM experiments that external forcings and related (e.g snow cover) feedbacks can

have an impact on SAT-GST coupling, with implications for borehole reconstructions, particularly at regional scales. The work developed herein implements a PPE setup and analysis on the same model ensemble. This allows for considering SAT-GST changes in simulations including a full configuration of natural and anthropogenic forcings and some single forcing simulations that can aid interpretation of the results. Differences between SAT and GST trends and between borehole reconstructions using SAT and GST generated BTPs allow for understanding about the limits of the second hypothesis stated above.

The CESM-LME has been reported to underestimate by 20% the 20th century trends (Otto-Bliesner et al., 2016). When considering the ensemble including all natural and anthropogenic forcings (ALL-F ensemble), linear fit GST (SAT) trends vary among ensemble members ranging between 0.28 and 0.51 (0.35 and 0.60) K century[-1]. This inter-simulation variability is an expected result of changing the initial conditions to generate the ensemble and reflects internal variability. We estimate trends during the 20th century as a metric of comparison of simulated and pseudo-reconstructed trends by considering: differences

between the final and initial 15-yr periods as well as linear trends; both approaches being consistent in delivering robust results. Results of the so-called idealized IBS PPE setup, in which sampling in time and space are not limited, produce a similar range





of GST trends as the simulations, thus supporting the overall performance of the SVD technique. The method is able to retrieve the long term trends through the LM, and the warming during the industrial period. Thus, the SVD approach itself renders reliable results in terms of retrieving the boundary GST signal. The method shows some sensitivity to increasing the number

of SVD modes. A number of modes consistent with previous modeling and experimental studies was selected here. Results are robust to small changes in this configuration.

A common result that affects all the subsequent tests in this work is that when methodological and physical constraints are imposed to the borehole reconstruction method, results depend on initial conditions and therefore on internal variability. This is a new although also arguably expected result, as imposing an specific spatial and temporal sampling setup at global

and regional scales may produce different effects on 20th century trends depending on the particular trajectory of internal variability and how effective the distribution of BTPs is in grasping the global and regional warming signals embedded within the range of internal variability. Our findings indicate that sampling can introduce detectable biases in borehole reconstructions both at global and regional scales. In the specific setup included herein, considering a realistic distribution of depths does not produce any detectable impact. However the distribution of logging dates and BTP locations does have some impact. At

global scales spatial and temporal masking introduces biases that can range between > 0 and 0.3 K century$^{-1}$ (GST - GST$_{mask}$ and IBS$_{L12}$ - B$_{mask}$ differences); spatial only masking reduces maximum differences to 0.1 K century$^{-1}$. This means that some simulations experience no significant change and in others masked PPE can experience underestimations of several tenths of a degree depending on the realization of internal variability. These are indeed small numbers but bear in mind that even a 0.1 K century$^{-1}$ change represents about 20% of the total warming in these simulations.

At regional scales, the impacts of temporal and spatial sampling vary across regions with differences (e.g. IBS$_{L12}$ - B$_{mask}$) that can range 0-0.2 K century$^{-1}$ in a relatively well sampled region as North America, to 0.4 K century$^{-1}$ in Europe, or 0.6 in the African domain. Most of it is due to the temporal sampling effect, particularly in the American and European regions where it gets reduced to values below 0.2 and 0.1 K century$^{-1}$, respectively, when considering spatial only masking; in the African domain spatial biases are larger and range between 0.1 and 0.3 K century$^{-1}$. At regional scales, some of these biases are larger

than their target temperature trend values. Thus, at regional scales spatial and temporal sampling is an issue. The effects are smaller over North America and larger over the other regions tested.

Temporal logging of the borehole records stands as the main sampling aspect contributing to the reduced skill in capturing the 20th century warming. This is because a large portion of the BTPs do not contain information of the warming during the last decades of the 20th century due to their relatively old logging dates. Such result suggests that the availability of

recent measurements highly influences the results of this type of temperature reconstruction. The continental scale analysis has provided further insight on this issue since the pseudo-reconstructed GST yields a generally good estimation of the simulated GST evolution during industrial times for areas with a relatively good distribution of recent BTPs measurements, as the North American continent. On the contrary, this accuracy is lost as the availability of recent BTPs is reduced, as for instance, over the African region. The temporal effects should be relatively small if the reconstructions are considered only up to the dates

of the oldest boreholes, at the expense of missing much of the warming developed during the last decades. Alternatively, strategies may be considered that would blend information from early borehole profiles with local instrumental data to mitigate





the missing trend effect (Harris and Chapman, 2001). Regarding the spatial sampling effects, the definition of the domains considered herein has been somewhat subjective. Perhaps more ad hoc domain setups can be specifically considered that reduce the effects of spatial sampling like in Africa. Otherwise, cases like the one selected in the African domain, including
very sparse sampling, should be avoided.

In the surrogate reality of the CESM-LME, the interpretation of the simulated SAT, derived from the reconstructed GST is additionally impacted by the physical processes that interrupt the long-term SAT-GST coupling. In the idealized scenario, the GST pseudo-reconstruction does not fully capture the global SAT increase during industrial times, missing about 20% of the simulated warming on average in the ALL-*F* ensemble. Globally, the larger increase of SAT relative to GST during industrial
times arises from the dominant influence of the GHG external forcing. Nevertheless, the contribution of individual forcings varies geographically as indicated by the regional analysis. While over the North American continent the overall response is similar than at the global scale, with a large contribution of the GHG forcing, over Europe and Africa the SAT-GST decoupling is dominated by the LULC forcing.

The impact of the long-term SAT-GST decoupling is superimposed on the limitations due to the methodological aspects. At
a global scale, the combined effect results in a biased representation of the simulated SAT 20th century trend by the realistic configuration, missing more than 50% of the simulated SAT (average values for the ensemble). Indeed, at this spatial scale, the largest contribution comes from the methodological aspects. Nonetheless, this may be different at smaller spatial scales. For instance, in North America, where the impact due to the methodological aspects is relatively small, the effect of the SAT-GST decoupling deteriorates the representation of the simulated SAT from the pseudo-reconstructed GST.

Overall, our results indicate that the combined effect of the methodological constraints and the physical SAT-GST decoupling leads to a systematic underestimation of the SAT 20th century trends at global scale; at regional scale overestimations can occur as in the African domain. Nonetheless, it is worth noting that despite this general pattern among the members of the ALL-*F* ensemble, there may be cases in which these impacts are relatively small or even disappear, consistent with previous studies (González-Rouco et al., 2006; García-García et al., 2016). This influence of internal variability and comparison with the actual
case in the real world can be better evaluated over reanalysis simulations of the 20th century (Hartmann et al., 2013), currently underway.

The results from PPEs herein are informative but not meant to be directly translated to the real-world cases. Nevertheless, they provide valuable information about the uncertainties of paleo-reconstructions in a controlled experimental framework (Smerdon, 2012). Despite the results of this work indicate a clear pattern in the potential sources of bias that can be found
in the borehole-based reconstruction, they should be interpreted with caution for real-world applications since several aspects may influence the level of impact. For instance, the use of other ESMs with different climate sensitivities, i.e. larger 20th century warming, and different representation of the influence of changes in land surface physics that could lead to a different representation of changes in the SAT-GST relationship. Additionally, the limitations in the local representation of sampling due to model resolution and more technical issues like the existence of local noise in BTPs have not been considered here.
In light of the results of this work, both the methodological issues and the physical biases of the borehole-based reconstructions would lead to an underestimation of the temperature increase from the LIA to present day, specially over the industrial period.



These findings are not able to explain the larger temperature increase suggested by the actual borehole estimations during the last centuries of the LM relative to those of some other proxy-based reconstructions.

*Author contributions.* This study is part of CMA's PhD. The experimental design was set up by CMA and JFGR. CMA carried out the data
processing, analyzed the results and wrote the paper. All of the other authors contributed to the analysis and discussion of the results and to writing the paper.

*Competing interests.* The authors declare that they have no conflict of interest.

*Acknowledgements.* This research was supported by an FPI grant (grant no. BES-2015-075019), from the Spanish Ministry of Economy, Industry and Competitiveness. We gratefully acknowledge the IlModels project, project no. CGL2014-59644-R. We also thank the
CESM1(CAM5) Last Millennium Ensemble Community Project and supercomputing resources provided by NSF/CISL/Yellowstone.



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
