# Peer review of "Methodological and physical biases in global to sub-continental borehole temperature reconstructions: an assessment from a pseudo-proxy perspective"

_Climate of the Past, 2019_

## Referee Comment (RC1) · Anonymous Referee #1 · 6 Nov 2019

Review of the manuscript "Methodological and physical biases in global to sub-continental borehole temperature reconstructions: an assessment from a pseudo-proxy perspective" by Camilo Melo-Aguilar, J. Fidel GonzaÌĄlez-Rouco, Elena GarcíÌĄa-Bustamante, Norman Steinert, Johann H. Jungclaus, Jorge Navarro, and Pedro J. Roldan-GoÌĄmez.

This manuscript assesses the spatial and temporal limitations of ground surface temperature histories reconstructed from borehole temperature profiles, as well as the effect of different forcings on the interpretation of those temperature histories. Authors

used a large ensemble of millennial simulations, which includes experiments to evaluate the role of individual forcings in the simulated climate evolution, for this assessment. I find this work relevant for the study of the Earth's system dynamics, paleoclimate, and climate modeling. The evaluation of limitations on the borehole database is of particular relevance. Nevertheless, I think some issues regarding the inversions and trend analysis should be addressed before the manuscript is ready for publication.

Inversions:

1- Authors limit the depth of the synthetic boreholes that they create mimicking the depth distribution of the measured borehole database. This limitation seems to be only applied to the bottom depth in the synthetic profiles, as the upper depth is not mentioned on the text. I assume, therefore, that the upper depth in the synthetic profiles is the surface. However, measured borehole temperature profiles rarely include data at the surface, and I wonder if authors have studied the case in which the upper depth of the synthetic profiles is configured as the upper measured depth in the corresponding borehole temperature profile. Would this have an effect on the results?

2- The authors aggregate inversions performed using profiles with different bottom depths. Previous studies [1,2] analyzing measured temperature profiles have shown that this practice biases the retrieved surface temperature histories, since the period of reference for each subsurface anomaly profile is different. I realize that authors are not using measured temperature profiles and that the synthetic profiles may not share this bias with the real case, but I wonder if authors have assessed the case in which all synthetic boreholes are truncated to a common bottom depth. Additionally, authors should include a brief note in the conclusions stating that although the depth masking do not affect their results, this is not the case when analyzing real borehole profiles.

3- Related to the previous comment, I have noticed that each simulated GST anomaly used to generate synthetic profiles have a different period of reference (lines 240-243). Therefore, the inversions of those synthetic profiles are relative to different climatolo-

gies. An easy solution to that would be to define a common period to compute all GST anomalies before generating the synthetic boreholes, and maintain such reference in the comparison with temperatures from the model.

4- It is very surprising that global mean temperatures from inversions computed using the B_mask configuration (green lines in Figs. 2, 3, 6, 7 and 8) are perfectly aligned to start in the year 2000 of the common era. Nonetheless, authors clearly state at several points on the text that the B_mask configuration considers the logging dates of the real borehole database. Does this mean that such different dates were not considered when aggregating the inversions? The green lines should display some kind of shift relative the ideal borehole scenario (IBS, red lines in Figs. 2, 3, 6, 7 and 8) configurations due to the different logging dates.

5- Which is the difference between GST_mask and B_mask? The authors state on line 306 that GST_mask was built by sampling GST in time, space and depth following the real borehole distribution. But the maximum simulated depth is 42m (35m is the last model node). I find this statement misleading, since borehole depths are much deeper than the simulated depth and it is not indicated which temperature is GST (although I suppose it is GST_L12, see Minor Concerns below).

Trend Analysis:

6- The comparison of trends under different masking configurations constitutes the core of the work, and yet I believe more details are needed regarding the trend analysis. There is no reference to the test applied to determine the significance of the trends. A common t-test would not be suitable for climate series due to the displayed autocorrelation, thus another test should be applied.

7- Also, it is not clear which is the method followed to generate the whisker plots comparing IBS_L12-GST and IBS_SAT-SAT trends in the linear fit case. SAT and GST temperatures are annual series, while the length of the time steps in IBS_L12 and IBS_SAT is 15 years. I guess that the trend of SAT (GST) and IBS_SAT (IBS_L12)

were estimated independently and then subtracted, but if so, I do not know if this is the correct approach. Should the annual temperatures be averaged in 15yr-periods, then subtract the reconstructions and estimate the trend of the resulting series? Maybe both approaches yield similar results, but a clarification here would be very helpful.

8- Crosses in Figs. 6 and 8 should represent the median of the GHG-only and LULC-only ensembles, since the authors are comparing against the median of the rest of ensembles (horizontal lines in the boxes). Note that the median is not always equal to the mean of the distribution.

9- In line 313, it is stated that "borehole reconstructions are able to retrieve the masked or unmasked GST" based on the results of Figure 2. However, there is no analysis of the trend of the GST-B_mask case. There is an analysis of the IBS_L12-B_mask case, but the IBS_L12 configuration is not the same as the unmasked case. Why not to include the trend analysis of the GST-B_mask case? Something similar can be said about the SAT-B_mask case in Figures 6 and 8.

Minor Concerns/Mistakes:

10- Equation 2 is incorrect. Right term of the equation should display the second order partial derivative of temperature. Check Carslaw and Jaeger (1959).

11- Section 3.2: does ST_L12 means GST_L12 as in the rest of the text? If so, please be consistent through the text.

12- Related to the previous comment, the definition of GST is not clear on the text since Section 4. Is it GST_L12?

13- Line 301: Which are the trends in Hartmann et al. (2013)? You could add those numbers to the text for an easier comparison with your results.

14- Line 381: "poor sampling enhances the influence of local behavior". What do authors mean here by local behavior? Please, expand this sentence.

15- Line 451: I believe authors mean "50% of the simulated trend".

16- Lines 308 and 461: which is the level of confidence in the statistical test applied to this trends?

17- Line 506: change "bu" by "by".

18- It is not clear which metrics are affected by the different masking configuration is Figs. 2 and 4. Is B_mask affected by those limitations or B_mask is always defined as indicated in Section 4.1? This should be easy to clarify on the captions and on the text.

References:

1- Beltrami, H., J.E. Smerdon, G. Matharoo, and N. Nickerson (2011). Impact of maximum borehole depths on inverted temperature histories in borehole paleoclimatology. Climate of the Past, 7, 745-756, 2011.

2- Beltrami, H., G. Matharoo, and J.E. Smerdon (2015). Impact of Borehole Depths on Reconstructed Estimates of Ground Surface Temperature Histories and Energy Storage. Journal of Geophysical Research - Earth Surface. 120(5): 763-778.

───────────────────────

---

## Referee Comment (RC2) · Anonymous Referee #2 · 18 Nov 2019

The paper uses a set of Earth System Model (ESM) simulations, with different initial conditions, to synthetically generate borehole subsurface temperature anomalies in in order to assess the methodological issues that may affect the reconstruction of ground surface temperature changes during the post-industrial period (1850-200 CE).

The premise here is that the ground energy content and its subsurface temperature field respond to the energy balance at the ground surface. Such balance is interplayed by a series of physical and biological processes, hydrology, hydrogeology, vegetation response, land use, etc. However if such physical and biological processes do not vary

over long time scales (decades to centuries), then the impulse response relation between surface air temperature and ground surface temperature should remain constant at long-time scales, thus surface air temperature changes should be reflected in ground surface temperatures mediated by an offset. By measuring vertical temperature profiles, subsurface temperature anomalies can be analyzed and interpreted as changes in the ground surface temperature as a response to surface temperature changes.

General Comments.

The paper is a meticulous set of synthetic experiments within a climate model space that uses the model reality to generate the data on which the experiments are carried out. That is, the input and output of the signals to be analyzed are know thus results are expected to be self-consistent and provide for the perfect pseudo reality in which experiments can be performed under controlled conditions.

The methodological approach for assessment of the impulse/response analysis are sound and well tested by the borehole reconstruction community and this paper represent a valuable contribution to the subject and to Climate of the Past.

I have several comments that I hope are useful. Some of my suggestions are outside the scope of the paper and I only mention them as suggestions for future work.

- The main issue for me, as the authors mention in the last part of the conclusions, is that all experiments are done in a noise-free environment. Data noise in borehole climatology is extremely important as it has an important effect on the maximum resolution that real data can provide. I assume that the authors are planning a second paper where the methodologies are explored in data and noise environment. I would encourage such paper.

-The authors examined the reconstructions based on (noise-free) subsurface temperature anomalies and not from complete borehole temperature profiles as these data are acquired.

Minor issues:

The paper is very dense requiring a lot of effort to keep up with the acronyms.

Line 36. A reference is needed for this claim.

Line 56-57. Depth of borehole temperature profiles was examined in Beltrami et al., 2011. A correction for logging time differential was used in Jaume-Santero et al., 2016.

Line 69: delete word "global"

Line 206. Convenient is misspelled.

Line 210: In the noise-free case presented here, the set up of the inversion depends on the choice of model, the geometry of the problem (i.e. the depth of the temperature profile and the depth sampling rate). The sampling rate is not given in the paper. I have assumed it was 1 m.

In practice, the number of eigenvalues retained in the inversion are also -besides the above mentioned factors - heavily determined by the noise level in the measurements. The tests with four and five eigenvectors retained thus do not have a straight forward meaning as in practical term the noise level would determining the number of principal components retained in the ground surface temperature reconstruction.

Section 3.2 (Pseudo) pseudo-proxy. The pseudo-proxy data generated here consist of subsurface temperature anomalies, not borehole temperature profiles (BTP). The BTP is a superposition of the subsurface temperature anomaly on the quasi-steady state temperate gradient. Perhaps authors should use BT anomaly (BTA).

Line 240 on: I am confused regarding the generation of the subsurface temperature anomaly field:

"Once the spatial distribution of the borehole network is represented in the CESM-LME grid, the LM STL12 series at each of these grid points is trimmed at the actual logging date according to the date distribution (Fig. 1a). Then, the temperature anomalies are

calculated with respect to the trimmed period mean" .

Does this mean that each "trimming period mean" or the "trimmed period" is used as a reference to estimate the anomalies? If so, in either case, each anomaly would have a different reference which could complicate the interpretation. In fact, they should take a single common period to estimate all anomalies for the comparison with the IBS case. Then verified that the differences on trimming period means may have something to do with the differences between the red and green inversion results in Figure 2 a.

In addition, how is the varying number of boreholes accounted for in the last 30-40 years for the green curve in Figure 2a?

Is the number of BT anomalies for IBS-L12 and B-mask the same in the most recent two inversion steps?

Do subsurface temperature anomalies in the IBS extend to the surface or to 7.8 m (node 12)? Real borehole temperature measurement standard analysis use data below 15-20 m?

Line 273: From Figure 2a, it is not clear to me that the temperature anomaly reconstructions capture the MCA-LIA transition. It seems to me that the resolution was lost by 1850 CE. The next sentence in line 273-274 seems to mention this but it seems contradictory.

Line 276: "Nevertheless, in model experiments that simulate larger MCA LIA changes the borehole reconstruction is able to recover somewhat warmer temperatures during the MCA (González-Rouco et al., 2006, 2009)." This would depend on the depth of the anomalies and also the number of principal components retained.

Line 346 and Lines 573-574 "The variability of the depth of the borehole records. . ." This is so only because the work is based on the analysis of subsurface temperature anomalies that contain little signals below 200 m because of the character of the ESM output used here.

Line 370 on, including Figure 3: The number of boreholes in Africa is small, and the area is huge. Much larger that the European slice in Fig 3b. Perhaps, giving the number of sites per unit area may help assess the discrepancies. The red and green lines in Fig 3c, seem contradictory for the cases shown. Are these differences arising from the different initial conditions of each of the 13 simulations? I wonder again whether the referencing over the trimmed period may have something to do with this (see comment on line 240.

I wonder what are the SAT-SAT mask) differences from the ESM simulations?

Line 632-633 : I would like the authors to expand in this issue. Perhaps, there is a need systematically collect additional borehole temperature profiles.

Summary and suggestions:

This is a good paper worthy of publication in COP.

Although out of the scope of this paper:

- It would be worth examining this problem for other ESM's simulations.

- I would also suggest that the authors consider writing a follow up paper with an identical analysis as in this paper, but based on a set of artificially generated full temperature logs, including simulated data noise. It may be that many of the differences that they observed in the noise-free set of experiments may change; and some differences could potentially be blurred significantly.

References:

Beltrami, H., J.E. Smerdon, G. Matharoo, and N. Nickerson (2011) Impact of maximum borehole depths on inverted temperature histories in borehole paleoclimatology, Clim. Past, 7, 745-756, 2011.

---

## Author Comment (AC1) · 17 Jan 2020

Camilo Melo-Aguilar1,2, J. Fidel González-Rouco1,2, Elena García-Bustamante3, Norman Steinert1,2, Jorge Navarro3, Johann H. Jungclaus4, and Pedro J. Roldan-Gómez1 1Universidad Complutense de Madrid, 28040 Madrid, Spain 2Instituto de Geociencias, Consejo Superior de Investigaciones Científicas-Universidad Complutense de Madrid, 28040 Madrid, Spain

3Centro de Investigaciones Energéticas, Medioambientales y Tecnológicas (CIEMAT), 28040 Madrid, Spain 4Max Planck Institut für Meteorologie, Hamburg, Germany

Correspondence: Camilo Melo-Aguilar (camelo@ucm.es)

The authors would like to thank the reviewers for their constructive suggestions and the time they devoted in reading and proof-reading the manuscript. We have tried to integrate all suggestions and think that the manuscript has improved with them. We do appreciate their contribution.

The next sections contain a detailed point by point response to the reviewers comments. Comments are labeled by re-

5 viewers and order of appearance, i.e. R2C3 is the third comment of reviewer 2. The original number by the reviewer is also preserved.

**1 Anonymous Referee 1**

GENERAL COMMENTS:

**10 R1C0: REVIEWER'S COMMENT:**

This manuscript assesses the spatial and temporal limitations of ground surface temperature histories reconstructed from borehole temperature profiles, as well as the effect of different forcings on the interpretation of those temperature histories. Authors used a large ensemble of millennial simulations, which includes experiments to evaluate the role of individual forcings in the simulated climate evolution, for this assessment. I find this work relevant for the study

of individual forcings in the simulated climate evolution, for this assessment. I find this work relevant for the study of the Earth's system dynamics, paleoclimate, and climate modeling. The evaluation of limitations on the borehole database is of particular relevance. Nevertheless, I think some issues regarding the inversions and trend analysis should be addressed before the manuscript is ready for publication.

**AUTHORS' RESPONSE:**

The authors welcome the positive perspective of the reviewer on the paper. We are grateful for the reviewer's

15

comments.

Please find below the comprehensive point-to-point response to your review.

**R1C1: REVIEWER'S COMMENT:**

5

Inversions: 1- Authors limit the depth of the synthetic boreholes that they create mimicking the depth distribution of the measured borehole database. This limitation seems to be only applied to the bottom depth in the synthetic profiles. as the upper depth is not mentioned on the text. I assume, therefore, that the upper depth in the synthetic profiles is the surface. However, measured borehole temperature profiles rarely include data at the surface, and I wonder if authors have studied the case in which the upper depth of the synthetic profiles is configured as the upper measured depth in the corresponding borehole temperature profile. Would this have an effect on the results?

**AUTHORS' RESPONSE: 10**

We appreciate that the reviewer noticed this issue that was not explained in the original text. The upper depth of the borehole temperature profiles (BTPs) was uniformly set to 20 m. We have included an explanation for this issue in the current version of the manuscript. Lines 202-203 of the manuscript has been modified as follow:

15  $...T_t(z)$  is evaluated at every 1 m depth interval up to a depth of 600 m in order to accommodate for the propagation of the LM surface temperature variations. Subsequently, the upper 20 m of the resulting BTPs are removed in order to avoid the influence of the annual signal and reproducing realistic depths of the water table (Jaume-Santero et al., 2016)."

**R1C2: REVIEWER'S COMMENT:**

- 20 2- The authors aggregate inversions performed using profiles with different bottom depths. Previous studies (Beltrami et al., 2011, 2015) analyzing measured temperature profiles have shown that this practice biases the retrieved surface temperature histories, since the period of reference for each subsurface anomaly profile is different. I realize that authors are not using measured temperature profiles and that the synthetic profiles may not share this bias with the real case, but I wonder if authors have assessed the case in which all synthetic boreholes are truncated to a common 25 bottom depth. Additionally, authors should include a brief note in the conclusions stating that although the depth masking do not affect their results, this is not the case when analyzing real borehole profiles

30

**AUTHORS' ANSWER:**

Figure 1 of this document illustrates the case in which all synthetic borehole temperature-anomaly profiles are truncated to a common bottom depth, following the reviewer suggestion, in the ALL- $F_2$  member of the ALL-F ensemble as an example. The global average profiles in the ideal borehole scenario using the standard configuration (600 m depth), as well as an alternative configuration in which all synthetic profiles are truncated at a "shallow" depth of 300 m are presented. In both cases, the upper 300 m show almost identical results. This is because in the surrogate reality of the model world, synthetic borehole temperature-anomaly profiles are directly created. Thus, the upper part of both profiles contains the same surface temperature signal of the las few hundred years. The shallow profile misses only the information of the earlier times that propagates deeper into the subsurface. The inversion of both profiles yields very similar results. Note that in both cases, the target temperature signal is accurately retrieved.

5

10

15

20

25

In the real-world cases, the anomaly profiles are obtained by subtracting the quasi-steady state parameters (i.e the geothermal gradient and equilibrium surface temperature, Beltrami et al., 2011). The latter is usually estimated from the bottom part of the borehole temperature profiles (BTPs) by linear fitting to the data and extrapolation to the surface. Therefore, truncating the BTPs at different depths may yield different values of the quasi-steady state parameters, and thus, of the temperature-anomaly profile impacting the results of inverted temperature histories. This source of uncertainty is present in experimental cases and has not yet been reproduced in pseudo-proxy experiments (PPEs). Therefore, in PPEs, the only source of uncertainty introduced by having borehole temperature-anomaly profiles of different depths is that related to the fact that shallower boreholes miss part of the past climate variability that deeper boreholes do not. This is as far as the approach in the text can reach and the differences are minor. Most likely, the loss of information of the earlier times in the shallower profile has a limited influence on the results because the CESM-LME simulations show a relatively low multi-centennial variability in the MCA-LIA transition.

We have included a mention stressing the fact that even though the synthetic borehole temperature-anomaly profiles are not strongly biased by the difference in the depth of the profiles, this may not be the case in real-world cases. Lines 341-344 of the manuscript has been modified as follow:

"...This implies that the effects of depth masking are negligible. Nonetheless, this may not be the case in real-world BTPs of different depths because anomaly profiles and elimination of the the background geothermal gradient is done by linear fitting at the bottom of the profile (Beltrami et al., 2011, 2015), thus introducing a source of uncertainty that has not yet been considered in PPE approaches."

**R1C3: REVIEWER'S COMMENT:**

30

3- Related to the previous comment, I have noticed that each simulated GST anomaly used to generate synthetic profiles have a different period of reference (lines 240-243). Therefore, the inversions of those synthetic profiles are relative to different climatologies. An easy solution to that would be to define a common period to compute all GST anomalies before generating the synthetic boreholes, and maintain such reference in the comparison with temperatures from the model.

**AUTHORS' ANSWER:**

5

20

We have implemented the approach of using a common period to compute all GST anomalies following the reviewer's suggestion. This is done for the ALL- $F_2$  member of the ALL-F ensemble as an example. All GST anomalies have been computed with respect to the 850-1960 CE mean. This period is selected since all borehole sites considered in this study are dated after 1960 CE. Once the anomalies are calculated with respect to a common period, each grid point anomaly (with the presence of borehole temperature profile) is trimmed at the actual logging date according to the real date distribution. Subsequently, the synthetic borehole temperature anomalies profile is created, and finally, the inversion using singular value decomposition is calculated. The period of reference is also used in the comparison with simulated GST as well as in the ideal scenario.

- Figure 2 herein show the results as it was presented in the manuscript (Fig 2a) as well the results from this alternative approach (Fig 2b). Note that the differences between these two approaches are hardly noticeable. In both cases, the B-mask inversion underestimate the ideal scenario (IBS). Other ensemble members yield comparable results. Therefore, the use of a different period of reference to estimate the GST anomalies, and subsequently, the borehole temperature-anomaly profiles cannot account for the differences between the IBS and the B-mask cases.
   We think that both are approaches correct. Using one or the other represents an appropriate adaptation of the
- real-world case. Since there are not important differences of using one or the other, we will keep the original approach in the document.

**R1C4: *REVIEWER'S COMMENT*:**

4- It is very surprising that global mean temperatures from inversions computed using the B-mask configuration (green lines in Figs. 2, 3, 6, 7 and 8) are perfectly aligned to start in the year 2000 of the common era. Nonetheless, authors clearly state at several points on the text that the B-mask configuration considers the logging dates of the real borehole database. Does this mean that such different dates were not considered when aggregating the inversions? The green lines should display some kind of shift relative the ideal borehole scenario (IBS, red lines in Figs. 2, 3, 6, 7 and 8) configurations due to the different logging dates.

**25 AUTHORS' ANSWER:**

This is an interesting issue raised by the reviewer. Indeed, as the reviewer points out, the B-mask pseudoreconstruction should be shifted relative to the ideal borehole scenario. The latter is because the most recent borehole temperature profiles considered in our study are dated in 2002 while the ESM simulations go up to 2005 CE.

30 We have changed the representation of the B-mask pseudo-reconstruction in all the figures along the document. Now, the shift of the B-mask relative to the IBS case is evident. See Fig 3 herein as an example. Note, however, that the differences with the figures presented in the original manuscript for the global case are very small. In addition, this has no influence in the estimation of the 20th century trends, which in the case of the B-mask inversion, have been estimated considering only the period 1900-last-available-date. We have included a mention on this issue in the text.

**R1C5: REVIEWER'S COMMENT:**

5- Which is the difference between GST-mask and B-mask? The authors state on line 306 that GST-mask was built by sampling GST in time, space and depth following the real borehole distribution. But the maximum simulated depth is 42m (35m is the last model node). I find this statement misleading, since borehole depths are much deeper than the simulated depth and it is not indicated which temperature is GST (although I suppose it is GST-L12, see Minor Concerns below).

**AUTHORS' ANSWER:**

10 We have corrected the description of GST-mask in this part of the document. Actually, the GST-mask is the masked version of GST with the actual borehole distribution only in space and time. As the reviewer pointed out, our explanation was misleading in the sense it is not possible to mask GST in depth because the maximum simulated depth is 42 m. The main message here is the analysis of the effect that decreasing spatial sampling with time would have on the representation of the global GST. We hope this correction makes the interpretation of this part clear. We have changed lines 306 and 321 of the manuscript. as follow:

"...Box-and-whisker plots are shown for all possible scenarios considered herein: GST,  $IBS_{L12}$ , GST masked with the realistic borehole configuration in space and time (GSTmask), Bmask as well as the differences among them (IBSL12 - GST, Bmask - GSTmask, GST - GSTmask and IBSL12 - Bmask)"

20

25

5

"...It is remarkable that when GSTs are masked, i.e. sampled, in space  $\frac{1}{2}$ , depth and time (GSTmask) following the real distribution, trends take a smaller range of values than those of GST."

In addition, we have included a table containing a detailed description of the different acronyms employed in the document following a suggestion of Reviewer #2. See R2C3 and Table 1 of this document.

**R1C6: REVIEWER'S COMMENT:**

Trend Analysis: 6- The comparison of trends under different masking configurations constitutes the core of the work, and yet I believe more details are needed regarding the trend analysis. There is no reference to the test applied to determine the significance of the trends. A common t-test would not be suitable for climate series due to the displayed autocorrelation, thus another test should be applied.

30

**AUTHORS' ANSWER:**

Indeed, an explanation of the test we applied to the significance of trends was not included in the original

manuscript. As the reviewer points out, in the case of temperature time series the regression residuals are not statistically independent and often depict strong autocorrelation. The effect of autocorrelation can be handled by considering an effective sample size in order to account for the non-independence of the residuals. This allows for estimating the significance of the trends considering both an adjusted standard error and adjusted degrees of freedom (Santer et al., 2000, 2008; Hartmann et al., 2013). In our case, we accounted for the temporal autocorrelation using a lag-1 autoregressive statistical model. Then, an effective sample size was calculated which is subsequently employed in the estimation of the standard error and degrees of freedom. Finally, the statistical significance of individual trends is obtained assuming that the statistic is distributed as Student's t. Santer et al. (2000, 2008) showed that the significant level computed from adjusted estimates of the standard deviation of regression residuals, the standard error and the t statistic, yield a more conservative estimation than without considering autocorrelation. Furthermore, the use of the effective sample size in the estimation of the critical tvalue result in an even more conservative approach.

For the statistical significance of trend differences we apply a "paired trends" test following Santer et al. (2008). The test statistic is of the form:

15
$$d = \frac{(b_1 - b_2)}{\sqrt{s(b_1)^2 + s(b_2)^2}}$$
(1)

Where  $b_1 - b_2$  represents the trend difference and  $s(b_1) + s(b_2)$  are the standard errors of  $b_1$  and  $b_2$ , respectively. The latter have been calculated using the effective sample size considering autocorrelation. For the statistically significance we applied a two-tailed test assuming that d is distributed as Student's t. The effective sample size is also considering to account for the reduced degrees of freedom (Storch and Zwiers, 1999).

We have included some changes in the manuscript in lines 309-313 and 323-324 as follow:

"...Interestingly, the frequency distribution of trends within the ALL-F ensemble is similar for both strategies (15-yr-diff and linear fit). The estimated global trends (GST in Fig. 3b) show statistically significant values (p < 0.05) that range between 0.3 and 0.6 K century-1 across the 13 simulations. Thus, internal variability has an impact in these trends 25 estimates. The significance of the trends is based on a t test and accounts for the temporal autocorrelation, using a lag-1 autoregressive statistical model, based on standard procedures for temperature time series (Santer et al., 2008; Hartmann et al., 2013). Likewise, autocorrelation is also accounted for the estimation of the reduced degrees of freedom (Storch and Zwiers, 1999). The trend values are somewhat smaller than those of the observational record that range between 0.73 and 0.83 K century-1 over the 1901-2012 period for the global mean surface temperature (Hartmann et al. 30 , 2013)."

"... The significance test (p < 0.05) of trend differences is based on a "paired trends" test following Santer et al., (2008) and also accounts for temporal autocorrelation. It is remarkable how the level of impact may depend on the interplay of

6

5

internal variability and BTP sampling."

**R1C7: REVIEWER'S COMMENT:**

5

15

20

7- Also, it is not clear which is the method followed to generate the whisker plots comparing IBS\_L12-GST and IBS\_SAT-SAT trends in the linear fit case. SAT and GST temperatures are annual series, while the length of the time steps in IBS\_L12 and IBS-SAT is 15 years. I guess that the trend of SAT (GST) and IBS-SAT (IBS\_L12) were estimated independently and then subtracted, but if so, I do not know if this is the correct approach. Should the annual temperatures be averaged in 15yr-periods, then subtract the reconstructions and estimate the trend of the resulting series? Maybe both approaches yield similar results, but a clarification here would be very helpful.

**10 AUTHORS' ANSWER:**

The box-and-whisker plots comparing the pseudo reconstructed IBS\_L12(IBS\_SAT) with the simulated GST(SAT) were generated by subtracting the individual trends following Santer et al. (2000, 2008). We have performed the analysis following the reviewer's suggestion in order to compare the results of both analyses. The annual time series have been averaged in 15 yr-periods and then subtracted from the pseudo-reconstructions. Then, the linear trend is estimated from the resulting series. Figure 4 of this document shows the results of such analysis for the IBS\_L12-GST case (methodological issues); similar results can be expected for the other cases. As the reviewer pointed out, both approaches yield similar results. Note that, even though there are small differences is some of the boxes compared to Fig. 2b of the original manuscript, the overall picture is in essence the same. The differences between IBS\_L12(B\_mask) and GST(GST\_mask) are distributed around 0 and GST(IBS\_L12) minus GST\_mask(B\_mask) are distributed around ~0.2 K century-1. The linear trends for GST and GST\_mask in Fig. 4 of this document have been estimated from the 15 yr average series. The resulting trends are slightly lower than the trends estimated from the annual time series, as in Fig. 2b of the original manuscript. Nonetheless, this has no impacts on the results. On the contrary, it shows that the trend estimates are robust regardless the differences between annual and 15 yr average time series.

25 Since the results from both approaches are similar, we keep the trend difference analysis by subtracting the individual trends, as it was originally presented. We have included an explanation of this in the text. See the caption of Fig 3 herein.

**R1C8: *REVIEWER'S COMMENT*:**

30

8- Crosses in Figs. 6 and 8 should represent the median of the GHG-only and LULConly ensembles, since the authors are comparing against the median of the rest of ensembles (horizontal lines in the boxes). Note that the median is not always equal to the mean of the distribution.

**AUTHORS' ANSWER:**

We are aware that the median and the mean does not necessarily coincide. However, due to the small sample size in the GHG- and LULC-only (only 3 ensembles members), we initially used the mean over the median in order to account for the information of the 3 ensemble member. The median on the other hand, would be representative of only one of them. Figures 5 and 6 of this document illustrate the effect of including the median instead of the mean for the single-f ensemble. Note that the differences with respect to Figs. 6 and 8 of the original document are small. Thus, using either the mean or the median yield identical results in terms of the dominant influence of one specific external forcing factor on the overall SAT-GST decoupling effect. Therefore, we have included the median of the GHG- and LULC-only ensembles in Figs. 6 and 8 following the reviewer's suggestion. Additionally, we have included some changes in the text related to this issue as follow:

10

5

"To provide a quantitative support to this statement the GHG- and LULC-only ensemble mean differences between the  $IBS_{SAT}$  -  $IBS_{L12}$  and  $IBS_{SAT}$  -  $B_{mask}$  20th century trends are included in Fig 6b (crosses and asterisks, respectively). the median of the differences between the  $IBS_{SAT}$  -  $IBS_{L12}$  and  $IBS_{SAT}$  -  $B_{mask}$  20th century trends in the GHG- and LULC-only ensemble are included in Fig 6b (crosses and asterisks, respectively). Results are almost identical if the

mean instead of the median of the single-*F* is included."

"...On the contrary, in the LULC-only ensemble the  $IBS_{SAT}$  -  $IBS_{L12}$  difference is negative and very small (-0.03 K Century-1) (-0.02 K Century-1) indicating a negligible contribution at this spatial scale."

20

15

"...Whereas in the GHG-only ensemble the mean median  $IBS_{SAT}$  -  $B_{mask}$  difference shows values in the same direction as the ALL-*F* ensemble, in the LULC-only ensemble the difference goes in the opposite direction (negative  $IBS_{SAT}$  -  $B_{mask}$  trend differences)."

**R1C9: REVIEWER'S COMMENT:**

9- In line 313, it is stated that "borehole reconstructions are able to retrieve the masked or unmasked GST" based on the results of Figure 2. However, there is no analysis of the trend of the GST-B\_mask case. There is an analysis of the IBS\_L12-B\_mask case, but the IBS\_L12 configuration is not the same as the unmasked case. Why not to include the trend analysis of the GST-B-mask case? Something similar can be said about the SAT-B\_mask case in Figures 6 and 8.

**30 AUTHORS' ANSWER:**

Figure 2 of the original manuscript shows that "borehole reconstructions are able to retrieve the masked or unmasked GST" since either IBS\_L12-GST or B\_mask-GST\_mask differences accurately retrieve the target signal evolution as they distributed around 0. Therefore, we use the IBS\_L12 pseudo-reconstruction as a reference GST for the comparison to the masked case, instead of the direct comparison with the simulated GSTs, in order to compare time series of the same type (i.e. 15-yr discretized pseudo-reconstructed time series). This was stated in Section 3.2 (lines 248-254 of the original manuscript). However, the GST-B\_mask comparison would yield similar results. Figure 7 of this document includes the box-and-whisker plot for the GST-B\_mask differences. Note that it shows a similar picture than the IBS\_L12-B\_mask column with some relatively small differences in the median (0.15 and 0.17 K century-1, respectively). Similar results can be expected in the SAT-B\_mask case. We think that, for the sake of consistency, it is better to keep the comparison between pseudo-reconstructed time series IBS\_L12(IBS\_SAT)-B\_mask. In addition, this has no effects on the overall results.

**SPECIFIC COMMENTS:**

10 Other minor points:

5

**R1C10 : 10- Equation 2 is incorrect. Right term of the equation should display the second order partial derivative of temperature. Check Carslaw and Jaeger (1959).**

Answer: We have corrected equation 2 accordingly.

15
$$\frac{\partial T}{\partial t} = \kappa \frac{\partial^2 T}{\partial z^2}$$
 (2)

**R1C11 : 11- Section 3.2: does ST\_L12 means GST\_L12 as in the rest of the text? If so, please be consistent through the text.**

20

25

Answer: ST\_L12 stands for the soil temperature at model layer 12 which is the soil layer we used as reference to create the synthetic BTP's as it is explained in Section 3.2. GST on the other hand, is defined as the temperature directly as the ground surface which in this case that would be the ST at the first model layer (L1; 0.007 m depth). Indeed, this ground surface temperature is the target signal of the borehole temperature reconstructions. Thus, in our study the pseudo-reconstructed GST, obtained from the IBSL12, is evaluated against the model GST defined here as the ST\_L1. We have included a definition of GST in order to make this issue more clear:

"We use the  $ST_{L12}$  as the reference GST ST to force the  $IBS_{L12}$  forward model because..."

"... In this work, GST, which is ultimately the target signal of the  $IBS_{L12}$ , is defined as the ST at the first soil layer (STL1; 0.007 m depth) following the same convention as in Melo-Aguilar et al., 2018)."

**R1C12 : 12- Related to the previous comment, the definition of GST is not clear on the text since Section 4. Is it GST\_L12?**

Answer: We have included a definition of GST. See response to R1C11 and also the response to R2C3.

**R1C13 : 13- Line 301: Which are the trends in Hartmann et al. (2013)? You could add those numbers to the text for an easier comparison with your results.**

Answer: The trends of the observational databases in Hartmann et al. (2013) have been included. Please note that the trends presented in Hartmann et al. (2013) represent global air surface temperature (not GST) for the 1901-2012 period since there is not available information for the GST at the global scale over this period of time. We have included the following:

"...The trend values are somewhat smaller than those of the observational record that range between 0.73 and 0.83 K century-1 over the 1901-2012 period for the global mean surface temperature (Hartmann et al., 2013)."

**R1C14 : 14- Line 381: "poor sampling enhances the influence of local behavior". What do authors mean here by local behavior? Please, expand this sentence.**

- Answer: Local behavior refers to the fact that obtaining the regional temperature average from only a few grid points is not totally representative of the whole region. Therefore, the regional mean may be biased by the small sample size (local behavior) relative to the whole region. We have included a short sentence clarifying this issue as follow:
- "...However, trends at these spatial scales can be highly dependent on internal variability and some simulations show no significant trends over the European and African domains. Additionally, poor spatial sampling enhances the influence of local behavior since only few grid points, often distributed within a relatively small area, determine the average of the whole region. Note that the representation of the 20th century trends in both the GSTmask and the Bmask spreads over a larger range than for North America, especially for the African region. This happens as a response to a decline in the availability of recent BTPs to calculate the regional averages during the last decades of the simulated period (see the maps in Fig 3.)"

**R1C15 : 15- Line 451: I believe authors mean "50% of the simulated trend".**

10

15

Answer: The reviewer is right, we actually refer to the 50% of the simulated SAT trend. We have included the ''missing word'' trend in the document.

**R1C16 : 16- Lines 308 and 461: which is the level of confidence in the statistical test applied to this trends?**

5

Answer: We have included the significance level in the statistical test of both individual trends and trend differences (p<0.05) in lines 308-309. However, it is worth noting that in line 461, we are not expressing statistically significance but only the fact that SAT minus GST trends over the mentioned areas are evidently larger in the GHG-only ensemble than in the All-F ensemble.

"... The significance test (p < 0.05) of trend differences is based on a "paired trends" test following Santer et al., (2008) and also accounts for temporal autocorrelation. It is remarkable how the level of impact may depend on the interplay of internal variability and BTP sampling."

**R1C17 : 17- Line 506: change "bu" by "by"**

Answer: It has been changed.

**15 R1C18 : 18- It is not clear which metrics are affected by the different masking configuration is Figs. 2 and 4. Is B\_mask affected by those limitations or B\_mask is always defined as indicated in Section 4.1? This should be easy to clarify on the captions and on the text.**

Answer: We agree that there may some confusion about the different masking configuration included in Figs. 2 and 4 since we tagged all of the masking cases as  $GST_{mask}(B_{mask})$ , even though, they include different masking configurations (i.e. spatial-only, spatial+depth and full spatial+depth+time). We have included a proper explanation of this issue in the manuscript and also the caption of Fig. 4. We hope that this explanation makes the different masking configuration more clear:

"...At this point, the question remains on what is the relative role of each of the three masking effects. Figure 3c addresses this by showing similar plots as the linear fit in Fig. 3b but considering only spatial and spatial plus depth masking. Note that even if the masked version is referenced only as GSTmask(Bmask) in the x-axis, however, in Fig. 3c-left(right) the masking includes spatial(spatial+depth) masking."

10

20

**2 Anonymous Referee 2**

**GENERAL COMMENTS:**

The authors would like to thank the reviewers for their constructive suggestions and the time they devoted in reading and proof-reading the manuscript. We have tried to integrate all suggestions and think that the manuscript has improved with them. We do appreciate their contribution.

The next sections contain a detailed point by point response to the reviewers comments. Comments are labeled by reviewers and order of appearance, i.e. R2C3 is the third comment of reviewer 2. The original number by the reviewer is also preserved.

**R2C0: REVIEWER'S COMMENT:**

10 The paper is a meticulous set of synthetic experiments within a climate model space that uses the model reality to generate the data on which the experiments are carried out. That is, the input and output of the signals to be analyzed are know thus results are expected to be self-consistent and provide for the perfect pseudo reality in which experiments can be performed under controlled conditions.

The methodological approach for assessment of the impulse/response analysis are sound and well tested by the borehole reconstruction community and this paper represent a valuable contribution to the subject and to Climate of the

noie reconstruction community and this paper represent a valuable contribution to the subject and to Cumate of the Past.

I have several comments that I hope are useful. Some of my suggestions are outside the scope of the paper and I only mention them as suggestions for future work.

**AUTHORS' RESPONSE:**

20 The authors welcome the positive perspective of the reviewer on the paper. We are grateful for the reviewer's comments. Please find below the comprehensive point-to-point response to your review.

**R2C1: REVIEWER'S COMMENT:**

- The main issue for me, as the authors mention in the last part of the conclusions, is that all experiments are done in a noise-free environment. Data noise in borehole climatology is extremely important as it has an important effect on the maximum resolution that real data can provide. I assume that the authors are planning a second paper where the methodologies are explored in data and noise environment. I would encourage such paper.

**AUTHORS' RESPONSE:**

We agree with the reviewer, our experiment represents an ideal noise-free environment that does not fully represent the real-world cases. As the reviewer points out, data noise in real borehole measurements is an important issue that required a carefully treatment of the data as well as a correct set up of the inversion. A follow up paper that additional consider noise in the experimental set up, as well as other factors not included in the this work, would be desirable. Up to date, the set up described in this paper is the most realistic adaptation

15

25

30

of the method to pseudo-proxy experiments. Nevertheless, it would be desirable to continue improving it in the future. We will explore the possibilities to develop such a work.

We have included a note in the last part of the conclusions stressing the fact that our experiment is developed in a noise free environment and that further consideration of this issue would be desirable in future works to fully represent the main features in which real-world borehole temperature reconstructions are developed.

5

"...Additionally, the limitations in the local representation of sampling due to model resolution and more technical issues like the existence of local noise in BTPs have not been considered here. Exploring these issues in future works would be desirable in order to have a more complete evaluation of the method. The latter would be specially interesting if new ensembles of LM simulations with ESMs including both All- and single-forcing experiments were developed in the frame of the CMIP6/PMIP4 (Coupled Model Intercomparison Project phase 6 / Paleoclimate Modeling Intercomparison Project phase 4; Eyring et al. , 2017; Jungclaus et al. , 2017, respectively). This would allow exploring the influence of different external forcing factors and different model physics that have some influence on, for instance, SAT-GST decoupling. Up to date, this issue can only be addressed with the use of the CESM-LME, as we have done in this study. Additionally, this work clearly supports the need for updating and expanding the borehole network. More and, if possible, deeper and good quality BTPs are needed."

15

20

10

**R2C2: REVIEWER'S COMMENT:**

-The authors examined the reconstructions based on (noise-free) subsurface temperature anomalies and not from complete borehole temperature profiles as these data are acquired.

**AUTHORS' RESPONSE**

We appreciate that the reviewer pointed out this issue. Indeed, the synthetic temperature profiles created from the surrogate reality of the model world represent only the transient perturbation induced by the past surface temperature variations. This transient component is superimposed on the background quasi-steady geothermal state. In real borehole temperature profiles (BTPs) the quasi-steady state component (geothermal gradient and equilibrium surface temperature) is usually estimated from the bottom part of the BTPs by linear fitting to the data and extrapolation to the surface. Then, the background components are subtracted from the BTPs to generate the temperature anomalies associated with the downwelling climatic components (Beltrami et al., 2011). In our case, we directly create temperature anomalies profile in a noise-free environment. We have included an explanation regarding this issue in the text (Section 3.1). See also the response to R2C1 and R2C9.

"The temperature at any depth z is The BTPs are determined by the combination of the geothermal heat flux, a reference ground temperature and the temperature perturbation  $T_t(z)$  induced by the surface temperature variations:

$$T(z) = T_0 + q_0 R(z) + T_t(z)$$
(3)

5

where  $q_0$  represents the surface heat flow density, R(z) is the thermal depth and  $T_0$  is a reference ground temperature. In the forward model, the quasi-steady state component  $(T_0 + q_0 R(z)t)$   $T_0 + q_0 R(z)$  can be set equal to 0 because the aim is to derive T(z) only as a function of the past surface temperature variations. The forward model, thus, determines the transient perturbation component  $T_t(z)$ , which can be thought as the anomaly with respect to the quasi-steady state thermal regime."

**10 SPECIFIC COMMENTS:**

Minor issues:

**R2C3 : The paper is very dense requiring a lot of effort to keep up with the acronyms.**

Answer: We have included a table containing a detailed description of the different acronyms employed in the document. See Table 1 (Table 4 in the corrected manuscripts). Additionally, we have included the following lines in the corrected manuscipt.

15

25

30

"...In order to allow for an easy identification of the different abbreviations and acronyms referenced along the document and their precise meaning, Table 4 contains a detailed description of them."

20 We hope this helps the reader to easily keep up with the different acronyms used in the paper.

**R2C4 Line 36: A reference is needed for this claim**

Answer: We have included the following references: Jansen et al. (2007); Fernández-Donado et al. (2013) in line 37 of the corrected manuscript.

**R2C5 *Line 56-57:* Depth of borehole temperature profiles was examined in Beltrami et al., 2011. A correction for logging time differential was used in Jaume-Santero et al., 2016.**

Answer: The point regarding the effect of the depth differences of borehole temperature profiles was also raised by Reviewer #1. Thus, we have included a mention regarding this issue. Please see the response to R1C2. For the logging time differential, actually, we used a similar approach to Jaume-Santero et al. (2016). In our case, global and regional averaging was also done on a yearly basis in order to account for the differences in logging dates.

**R2C6 Line 69: Delete word "global"**

Answer: The word "global" has been deleted.

**R2C7 Line 206: Convenient is misspelled**

Answer: It has been corrected.

5 R2C8 *Line 210:* In the noise-free case presented here, the set up of the inversion depends on the choice of model, the geometry of the problem (i.e. the depth of the temperature profile and the depth sampling rate). The sampling rate is not given in the paper. I have assumed it was 1 m.

In practice, the number of eigenvalues retained in the inversion are also -besides the above mentioned factors heavily determined by the noise level in the measurements. The tests with four and five eigenvectors retained thus do not have a straight forward meaning as in practical term the noise level would determining the number of principal components retained in the ground surface temperature reconstruction.

Answer: The sampling rate is indeed 1 m, as it was stated in Section 3.1, line 198 of the original manuscript. Our PPE is indeed developed in a noise-free environment as in the case of previous PPEs (e.g. González-Rouco et al., 2009; García-García et al., 2016). In real-world cases the level of noise plays a relevant role on the retained number of eigenvectors. Higher noise limits the number of eigenvectors retained, and thus, the resolution of the retrieved climatic signal (Beltrami and Bourlon, 2004). We have opted for a configuration with a conservative low number of eigenvalues (3) even if this is a noise-free PPE, and actually find consistence between the inverted and the simulated trends. We have additionally shown results for 4 and 5 eigenvectors, as these numbers have been used in previous experimental (Beltrami and Bourlon, 2004; Jaume-Santero et al., 2016) and PPEs (González-Rouco et al., 2009) works. We have included a mention regarding this issue in the text (lines 354-356) as follow:

20

25

10

15

"...In real-world data the level of noise in BTPs limits the number of retained principal components (Beltrami and Bourlon, 2004). We have used here a conservative low value that may be consistent with real-world applications including noise."

R2C9 Section 3.2: Section 3.2 (Pseudo) pseudo-proxy. The pseudo-proxy data generated here consist of subsurface temperature anomalies, not borehole temperature profiles (BTP). The BTP is a superposition of the subsurface temperature anomaly on the quasi-steady state temperate gradient. Perhaps authors should use BT anomaly
 (BTA).

Answer: The synthetic temperature profiles we create are indeed only the temperature perturbation induced by the surface temperature variations. See the response to as well as the response to R2C2. We have included an "...using LM surface temperature annual anomalies from the CESM-LME as the upper boundary condition. Although the synthetic temperature profile represent only the transient perturbation component,  $T_t(z)$ , of the BTP, it will be denoted as BTP thorough the document to avoid confusion."

- R2C10 Line 240: Line 240 on: I am confused regarding the generation of the subsurface temperature anomaly field: "Once the spatial distribution of the borehole network is represented in the CESM-LME grid, the LM STL12 series at each of these grid points is trimmed at the actual logging date according to the date distribution (Fig.
- 10 1a). Then, the temperature anomalies are calculated with respect to the trimmed period mean".
  Does this mean that each "trimming period mean" or the "trimmed period" is used as a reference to estimate the anomalies? If so, in either case, each anomaly would have a different reference which could complicate the interpretation. In fact, they should take a single common period to estimate all anomalies for the comparison with the IBS case. Then verified that the differences on trimming period means may have something to do with the differences between the red and green inversion results in Figure 2a.
  - In addition, how is the varying number of boreholes accounted for in the last 30-40 years for the green curve in Figure 2a?

Is the number of BT anomalies for IBS-L12 and B-mask the same in the most recent two inversion steps?

Do subsurface temperature anomalies in the IBS extend to the surface or to 7.8 m (node 12)? Real borehole temperature measurement standard analysis use data below 15-20 m?

Answer: Reviewer #1 expressed a similar concern. We have performed the test taking a common period to estimated the anomalies. Figure 2 of this document shows the results of such test. Note that it yields almost identical results to the estimation of trends using the trimmed period as we presented in the manuscript. Thus, the differences between the ideal scenario (red line) and the B-mask (green line) cannot be explained by the differences on trimming periods. For further details please see the response to R1C3 for details.

- About the varying number of boreholes accounted for in the last 30-40 years, the global and regional averaging was done on a yearly basis in order to account for the differences in logging dates. Therefore, the number of BT anomalies for IBS-L12 and B-mask is not the same in the most recent two inversion steps. A similar approach has been used in other works that make use of actual borehole temperature profiles measurements (e.g. Jaume-Santero et al., 2016). For further details, see the response to R1C4.
- Finally, the subsurface temperature anomalies in the IBS cases start at 20 m depth. Actually, this was not originally explained in the document. Thus, we have included a proper explanation in the text. See also the response to R1C1.

5

20

" $T_t(z)$  is evaluated at every 1 m depth interval up to a depth of 600 m in order to accommodate for the propagation of the LM surface temperature variations. Subsequently, the upper 20 m of the resulting BTPs are removed in order to avoid the influence of the annual signal and reproducing realistic depths of the water table (Jaume-Santero et al., 2016)."

**5 R2C11 *Line 273:* Line 273: From Figure 2a, it is not clear to me that the temperature anomaly reconstructions capture the MCA-LIA transition. It seems to me that the resolution was lost by 1850 CE. The next sentence in line 273-274 seems to mention this but it seems contradictory.**

Answer: The pseudo-reconstructed GST from the IBS shows, indeed, a nearly flat transition from the MCA to the LIA. This is in part due to the low multi-centennial variability depicted by the simulated GST itself. The simulated GST shows non-significant negative trends for the period 850-1850 CE. We have modified this statement in the text accordingly:

"...In general, the  $IBS_{L12}$  pseudo-reconstruction reasonably reproduces the gross features of the low-frequency GST variations over the LM in the CESM-LME. For instance, the transition from the MCA to a colder LIA a small non-significant multi-centennial cooling from the MCA to the LIA can be detected and the warming over industrial times are is successfully captured in both cases."

**R2C12 Line 276: Line 276: "Nevertheless, in model experiments that simulate larger MCA LIA changes the borehole reconstruction is able to recover somewhat warmer temperatures during the MCA (González-Rouco et al., 2006, 2009)." This would depend on the depth of the anomalies and also the number of principal components retained.**

Answer: As the reviewer points out, the number of principal components retained in the solution play also an important role in the resolution of the retrieve surface temperature histories. Indeed, this is evident in the results of González-Rouco et al. (2009). We have included a mention on this issue in the text as follow:

25

10

15

"...Nevertheless, in model experiments that simulate larger MCA-LIA changes the borehole reconstruction is able to recover somewhat warmer temperatures during the MCA if the depth of the anomalies and the number of eigenvalues retained is adequate. (González-Rouco et al., 2006, 2009)."

30 R2C13 *Line 346:* Line 346 and Lines 573-574 "The variability of the depth of the borehole records..." This is so only because the work is based on the analysis of subsurface temperature anomalies that contain little signals below 200 m because of the character of the ESM output used here.

Answer: We have included a sentence stressing this fact in the conclusions.

...Our findings indicate that sampling can introduce detectable biases in borehole reconstructions both at global and regional scales. In the specific setup included herein, considering a realistic distribution of depths does not produce any detectable impact. This may be, in part, due to the little signal contained in the synthetic BTPs below 200-300 m depth because the CESM-LME simulations depict relatively low multi-centennial surface temperature variability.

R2C14 Line 370: Line 370 on, including Figure 3: The number of boreholes in Africa is small, and the area is huge. Much larger that the European slice in Fig 3b. Perhaps, giving the number of sites per unit area may help assess the discrepancies. The red and green lines in Fig 3c, seem contradictory for the cases shown. Are these differences arising from the different initial conditions of each of the 13 simulations? I wonder again whether the referencing over the trimmed period may have something to do with this (see comment on line 240)

Answer: There is indeed a relatively low coverage of borehole grid points with respect to the total grid point of the regions we considered (N. America=106/488, Europe=33/191 and Africa=37/215), although the ratio 15 for Africa is comparable to that of Europe. The effect related to this issue was stated in lines 381-384 of the original document: "poor spatial sampling enhances the influence of local behavior". Besides the ratio of borehole grid points relative to the total of grid points within each of the regions, the spatial distribution of the borehole sampling is also an important factor. The latter is specially relevant in the most recent decades when the number of available borehole-grid points decreases significantly. Furthermore, these recent borehole logs tend to be cluster over some specific areas, which enhance the local emphasis of "poor spatial sampling". This is particularly the case of the African region where there are only five borehole sites dated after 1990 CE; four of them located in the southern part of the region. In the manuscript, there is a sentence stressing this fact (lines 381-384 in the original document). The spatial sampling, and the decrease of available borehole sites with time, can be inferred from the maps in Fig. 3. We have included a note in the text to draw the attention on this issue. See the response to R1C14. Regarding the case example shown in Fig 3c, we chose a case for each of the regions that represents the median of the distribution in the boxplots. Note that for the African case, the 20th century trends of the red and the green lines are in agreement with the median value shown in the boxplots. An increasing trend of ~0.2 K century-1 depicted by the red line and a decreasing trend of  $\sim -0.1$  K century-1 depicted by the green one. Such difference in the 20th century trends arises from the poor sampling in the last decades to calculate the regional average in the 30 Bmask case (green line) as explained above.

5

10

20

25

**R2C15 : I wonder what are the SAT-SAT mask) differences from the ESM simulations?**

Answer: Figure 8 of this document includes the SAT-SATmask differences. We have included the SAT masked using both the spatial-only mask and the spatial+temporal mask to allow comparison of the effects from the different masking configurations. Note that in both cases the masking leads to an overall underestimation of

the global SAT. This is comparable to the effect of masking GST with the spatial-only and the spatial+temporal masks as shown in Fig. 2 of the manuscript, although in the case of SAT the effect is slightly smaller. The SAT-SATmask trend differences considering spatial-only(spatial+temporal) masking are centered around 0.075(0.077) K century-1 (Fig 8 of this document). However, in the spatial+temporal masked case there is a larger spread of SAT-SATmask differences across the 13 member of the ALL-F ensemble, ranging from -0.143 to 0.185 K century-1. The latter is due to the enhanced effect of internal variability produced by temporal sampling. Even the spatial sampling alone can produce differences of almost 0.16 K century-1 over a trend of 0.5 K century-1.

**R2C16 Line 632-633: Line 632-633 : I would like the authors to expand in this issue. Perhaps, there is a need systematically collect additional borehole temperature profiles.**

- 10 Answer: We have included a mention on this issue in the conclusions following the reviewer's suggestion, stressing the fact that borehole temperature reconstructions would be benefited from systematically collecting additional borehole temperature profiles in the future. We understand the logistic and funding challenges, but it would be really useful.
- "...Alternatively, strategies may be considered that would blend information from early borehole profiles with local 15 instrumental data to mitigate the missing trend effect (, Harris and Chapman, 2001). In addition, this type of analysis would benefit from re-logging boreholes whenever possible and logging additional BTPs in the future; thus updating the network."
- 20 "... Additionally, this work clearly supports the need for updating and expanding the borehole network. More and, if possible, deeper and good quality BTPs are needed.

In addition, se the response to R2C1

25 R2C17 Summary and suggestions:

This is a good paper worthy of publication in COP.

Although out of the scope of this paper:

- It would be worth examining this problem for other ESM's simulations.

- I would also suggest that the authors consider writing a follow up paper with an identical analysis as in this

paper, but based on a set of artificially generated full temperature logs, including simulated data noise. It may 30 be that many of the differences that they observed in the noise-free set of experiments may change; and some differences could potentially be blurred significantly.

Answer: We appreciate the reviewer considers our work is worth for publication in COP. We agree that examining both the methodological and physical aspects on borehole temperature reconstructions we addressed in this study, using other ESM's simulation would yield valuable additional information on this topic. In fact, The CMIP6/PMIP4 (Eyring et al., 2017; Jungclaus et al., 2017) experiments offer an opportunity to address this issue with state-of-the-art ESM. The latter would be specially interesting if new ensembles of simulations with ESMs including both All- and single-forcing experiment were developed, thus allowing to explore the influence of different external forcing factors on the physical-related processes as we did in the present work. Up to date, this is only possible by using the Community Earth System Model-Last Millennium Ensemble (Otto-Bliesner et al., 2016). Extending this approach to other ESM would yield additional information of the influence of different external forcing factors and different model physics on, for instance, the long-term SAT-GST relationship. It would also help to consider cases of model having a larger temperature response (i.e. climate sensitivity), thus gaining more confidence on the overall effects.

A follow up paper that considers additional factors in the experimental set up, would also be desirable. We will explore the possibilities to develop such a work. We have included some of these considerations in last part of the conclusions as follow:

..."Exploring these issues in future works would be desirable in order to have a more complete evaluation of the method. The latter would be specially interesting if new ensembles of LM simulations with ESMs including both Alland single-forcing experiments were developed in the frame of the CMIP6/PMIP4 (Coupled Model Intercomparison Project phase 6 / Paleoclimate Modeling Intercomparison Project phase 4; Eyring et al. , 2017; Jungclaus et al. , 2017, respectively). This would allow exploring the influence of different external forcing factors and different model physics that have some influence on, for instance, SAT-GST decoupling. Up to date, this issue can only be addressed with the use of the CESM-LME, as we have done in this study. Additionally, this work clearly supports the need for updating and expanding the borehole network. More and, if possible, deeper and good quality BTPs are needed."

5

10

| Acronym             | Meaning                                                                                                  |
|---------------------|----------------------------------------------------------------------------------------------------------|
| GST                 | Simulated ground surface temperature defined as the first soil layer temperature (ST L1 ;     |
|                     | 0.007 m depth)                                                                                           |
| SAT                 | Simulated 2 m air temperature. Original model output TREFHT                                              |
| IBS L12  | Ideal borehole scenario created from $ST_{\rm L12}$ as the boundary condition for the forward            |
|                     | model                                                                                                    |
| IBS SAT  | Ideal borehole scenario created from SAT as the boundary condition for the forward                       |
|                     | model                                                                                                    |
| GST mask | GST masked with the realistic representation of the variability of the spatio-temporal                   |
|                     | distribution of the global borehole network. In some cases $\mbox{GST}_{\mbox{mask}}$ refers to the full |
|                     | spatial+date sampling whereas in other cases it refers only to spatial sampling (i.e. Fig                |
|                     | 4)                                                                                                       |
| B mask   | Realistic scenario of the borehole temperature inversions including sampling in space,                   |
|                     | time and depth. It may also refer to the cases in which the sampling is only in space or                 |
|                     | space+depth (i.e. Figs 2 and 4)                                                                          |

**Figure 1.** Left: global average of the synthetic borehole temperature anomalies profiles from the ideal borehole scenario (IBS) using a bottom truncating depth of 600 and 300 m for comparison (IBS600 and IBS300, respectively). Results are shown for the ALL- $F_2$  member of the ALL-F ensemble as an example. Right: LM global GST annual anomalies and the corresponding 31-yr filtered outputs, the global IBS600 and the IBS300 pseudo-reconstructions for the ALL- $F_2$  realization. Note the different discretization in the x axis after 1700 CE. A zoom of the last 205 year is shown to allow visualization. The dashed lines depict the linear trends for the 1900-2005 CE period. The values are indicated on the right-hand side in K century-1.